# Review on the Genus *Stylophoronychus* (Acari: Tetranychidae), with Description of a New Species [note 1]

**DOI:** 10.3390/insects13121176

**Published:** 2022-12-19

**Authors:** Xiaojuan Pan, Ronald Ochoa, Daochao Jin, Tianci Yi

**Affiliations:** 1Institute of Entomology, Guizhou University, Guiyang 550025, China; 2The Guizhou Provincial Key Laboratory for Plant Pest Management of Mountainous Region, Guiyang 550025, China; 3The Scientific Observing and Experimental Station of Crop Pest in Guiyang, Ministry of Agriculture P. R. China, Guiyang 550025, China; 4Systematic Entomology Laboratory, ARS, USDA, Beltsville, MD 20705, USA

**Keywords:** *Stylophoronychus*, ontogenetic development, new species, synonym, *Bambusa* spp., spider mites

## Abstract

**Simple Summary:**

The spider mite family Tetranychidae includes 85 genera and more than 1300 species worldwide, and is the group of phytophagous mites that has the greatest impact on the agricultural economy. *Stylophoronychus* is a small genus of six species that belongs to the tribe Aponychini which was reinstated as a valid taxon containing three genera (*Aponychus*, *Paraponychus* and *Stylophoronychus*). Here, a new species *Stylophoronychus wangae*
**Pan, Jin & Yi sp. nov.** from Majiang Country, Guizhou Province, China (the Oriental realm) is described based on the deutonymph and adults. Two species, *S. guangzhouensis* (Ma and Yuan, 1980) and *S*. *lalli* (Prasad, 1975) are considered junior synonyms of *S. vannus* (Rimando, 1968). The taxonomy of the genus *Stylophoronychus* is reviewed and the studies on its ontogenetic development are discussed.

**Abstract:**

Only two species of the genus *Stylophoronychus, S*. *baghensis* (Prasad, 1975) and *S*. *guangzhouensis* (Ma and Yuan, 1980), have been recorded in China. Herein we describe a new species *Stylophoronychus wangae*
**Pan, Jin & Yi sp. nov.** based on characteristics of the deutonymphs and adults. The synonym of *S. guangzhouensis* (Ma and Yuan, 1980) and *S*. *lalli* (Prasad, 1975) with *S. vannus* (Rimando, 1968) is proposed. A redescription of *S. vannus* (Rimando, 1968) based on the adults of both sexes, deutonymphs and a protonymph is given. The ontogenetic changes of leg chaetotaxy in two species are given and discussed. The updated key to the species of *Stylophoronychus* of the world is provided.

## 1. Introduction

The genus *Stylophoronychus* belongs to the tribe Aponychini of the subfamily Tetranychinae and contains six species (*S. baghensis*, *S*. *guangzhouensis*, *S*. *insularis*, *S*. *lalli*, *S*. *nakaoi* and *S*. *vannus*) [1,2,3,4]. Most of these species are distributed in the Oriental realm, primarily found on *Bambusa* spp., with the exception of *S. insularis* which has been reported only from Araliaceae sp. in the Ethiopian realm [1,3,5]. *Stylophoronychus* was originally erected as the subgenus of *Aponychus* by Prasad to accommodate *S. baghensis* based on two strong lobes of the stylophore [1,6]. Later, the subgenus was elevated by Meyer [7] to the generic status due to the key character of nine pairs of dorsal setae on the hysterosoma, and all the species of *Stylophoronychus* were reclassified based on this character [1,7]. For more detailed changes in the classification of the genus refer to Zhang et al. [1].

*Stylophoronychus guangzhouensis* is endemic to China and resembles *S. vannus* and *S. lalli* [1,2,5,8,9]. Hernandes and Feres [2] compared three species of *Stylophoronychus* and one species of *Aponychus.* Zhang et al. [1] conducted a detailed comparative analysis and provided a key to the species of *Stylophoronychus*. They considered *A. bambusae* a junior synonym of *S. vannus* while *S. guangzhouensis* (not examined), *S. lalli* and *S. vannus* were separate and valid species. There is a slight difference in the shape of setae *h*_1_ being more palmate in *S. vannus* in comparison to the relatively slimmer shape observed in *S. lalli* and *S. guangzhouensis.* After performing a comparative analysis of the morphological characters of five species (*S*. *baghensis*, *S*. *guangzhouensis*, *S*. *lalli*, *S. vannus* and *S. wangae*
**Pan, Jin & Yi sp. nov.**) of the genus (not *S*. *insularis* and *S*. *nakaoi),* we consider *S*. *guangzhouensis* and *S*. *lalli* to be junior synonyms of *S. vannus.* We examined Ma’s collection of *S. guangzhouensis* (originally deposited at the Shanghai Museum of Natural History, Shanghai, China and Shanghai Agricultural College, Shanghai, China) which included 13 females, six males, and five deutonymphs from Jinghong City, Yunnan Province. We also examined new collections of *S. guangzhouensis* collected by Tian-Ci Yi from leaves of *Chimonobambusa quadrangularis* (Fenzi) Makino in Guangzhou City, Guangdong Province which included one protonymph, one deutonymph and six females. In addition, specimens of *S. vannus* from six collections that include one paratype female, two females, three males and one paratype female of *S*. *lalli*, and two paratype females of *S*. *baghensis* borrowed from the USNM, were compared and studied. 

The ontogenetic development of leg chaetotaxy in spider mites having one pair of duplex setae on tarsus I, which is rare, is poorly known. To reach a better understanding of this genus, a protonymph, deutonymphs and adults of *S. vannus* and a deutonymph and adults of *S. wangae*
**sp. nov.** were collected and reared to examine the ontogenetic development and provide a description of the immature stages of these species.

## 2. Materials and Methods

The mite specimens studied were examined using a Leica DM 5000B microscope with differential interference contrast. Line drawings were prepared with the aid of a drawing tube attached to the microscope. Photographs were taken under oil immersion using a camera (Nikon DS-Ri 2) attached to the microscope (Nikon Ni E). Measurements were obtained using software (Nikon NIS Elements AR 4.50) and are provided in micrometers (µm). Length of the idiosoma was measured from the center of the setal base of *v*_2_ to *h*_1_, while width was measured from the center of the setal base of *c*_1_ to *c*_3._ The measurements are presented for the holotype followed by the range of paratypes in parentheses. Morphological terminology follows that of Lindquist [10]. 

## 3. Results

### 3.1. Taxonomic Discussion in S. guangzhouensis, S. lalli, S. vannus

These three species were compared based on the following characteristics: 

(1) Idiosoma. The body shapes of the adult females for all three species have the idiosoma nearly oblong, slightly longer than wide, and margins on both sides approximately parallel (Figure 1).

(2) Dorsal setae. For the three species shown in Figure 1 and Table 1, the dorsocentral setae are long, linear, inserted on tubercles and pubescent. They are characterized such that their length, with the exception of the first row, is approximately equal or shorter than distances between their bases. The key provided by Zhang et al. [1] uses the relative length of *e*_1_ subequal and less than the distance between their bases to distinguish among the three species, but this character is unreliable. According to Hernandes and Feres [2], the shape of setae *h*_1_ is the only character that can distinguish these three species reliably but only when a large series of specimens is examined. However, after examining 19 female specimens of *S. guangzhouensis,* we found the shape of setae *h*_1_ is variable, ranging between spatulate, fan-like or palmate. Figure 2 and Figure 3 show that *S. guangzhouensis* shares the shapes of *h*_1_ setae with the other two species. This variation of the shape of *h*_1_ is most likely intraspecific polymorphism and cannot be used as a character to distinguish the three species.

(3) Patterns of hysterosomal median protuberance. *S. guangzhouensis*, *S. lalli* and *S. vannus* have a protuberance on the hysterosomal median that arched upward, shaped like the outline of butterfly wings, densely covered with irregular patterns of circles or the fusion of many circles (Figure 4). 

(4) Patterns of integument on medial prodorsum. *S. guangzhouensis*, *S. lalli* and *S. vannus* have a nearly square bulge on the prodorsal median area with a conspicuous pattern of highly wrinkled ornamentation that resembles a brain cortex (described by Hernandes and Feres) (Figure 5). 

(5) Stylophore. There are similar apicodorsal granulations on the two strong lobes of the stylophore with a median convex area in *S. guangzhouensis*, *S. lalli* and *S. vannus* (Figure 6). 

(6) Leg chaetotaxy. Females of the three species have the same legs chaetotaxy I–IV (eupathidia and solenidia in parentheses): trochanters 1-1-1-1; femora 6-5-3-1; genua 1-1-1-1; tibiae 3(1)-1-1-1; tarsi 7(3)(2)-7(3)(1)-6-6(0)(1). However, the leg chaetotaxy shows an intraspecific polymorphism for the three species. Three females of *S. vannus* were examined, and one paratype female shows 6-6(1) on tarsus III–IV, while the other two females show 7-7(1). Leg chaetotaxy of tarsus III–IV in *S. guangzhouensis* is more variable.

Based on the above common typical characteristics analysis, *S*. *guangzhouensis*, *S*. *lalli* and *S. vannus* are considered to be synonymous, the latter species having priority. The variations on tarsus III–IV of *Stylophoronychus vannus* as shown in Table 2.

### 3.2. Redescription of S. vannus

Family Tetranychidae Donnadieu

Subfamily Tetranychinae Berlese

Tribe Aponychini Rimando & Corpuz-Raros

Genus *Stylophoronychus* Prasad, 1975

*Aponychus vannus* Rimando, 1968

*Aponychus* (*Stylophoronychus*) *vannus* (Rimando); Prasad, 1975

*Stylophoronychus vannus* (Rimando); Meyer, 1987


**Redescription**


Figure 7, Figure 8, Figure 9, Figure 10, Figure 11, Figure 12, Figure 13, Figure 14, Figure 15, Figure 16, Figure 17, Figure 18 and Figure 19

Adult female (*n* = 23) 

Dorsum (Figure 7 and Figure 8). Idiosoma vermilion, 296–338 long, 264–297 wide, nearly oblong, slightly longer than wide and lateral margins approximately parallel. Dorsal setae white. Hysterosoma with two distinct protuberances anterior, one small and bears seta *c*_1_, posterior, one large constricted laterally (between *d*_1_ and *e*_1_), and bears setae *d*_1_ and *e*_1_. 

Prodorsum with three pairs of palmate to spatulate setae, covered with short longitudinally aligned spinules, *v*_2_ 31–39, *sc*_1_ 30–40 and *sc*_2_ 44–56. Distances between setal bases: *v*_2_–*v*_2_ 79–87, *sc*_1_–*sc*_1_ 111–127, *sc*_2_–*sc*_2_ 224–239. Prodorsum with a pattern resulting in highly wrinkled ornamentation on those transverse ridges.

Hysterosoma with 11 pairs of setae (*c*_1–3_, *d*_1–2_, *e*_1–2_, *f*_2_, *h*_1–3_), *f*_1_ absent and *f*_2_ marginally positioned. Setae *c*_1_, *c*_3_, *d*_1_ and *e*_1_ elongate palmate to long linear, *c*_2_ much shorter than other dorsal setae, about as long as one third of seta *c*_1_; *e*_2_, *f*_2_ and *h*_1_ palmate or spatulate. Setae *h*_2–3_ of differing morphology, similar to other ventral setae and inserted posteroventrally. Seta *c*_1_ slightly longer than the distance to setae *d*_1_. Seta *d*_1_ shorter than the distance to seta *e*_1_. Length of setae: *c*_1_ 58–78, *c*_2_ 24–29, *c*_3_ 50–65, *d*_1_ 60–76, *d*_2_ 34–51, *e*_1_ 59–72, *e*_2_ 43–48, *f*_2_ 40–47, *h*_1_ 37–42, *h*_2_ 18–27, *h*_3_ 15–25. Distances between setal bases: *c*_1_–*c*_1_ 39–53, *c*_2_–*c*_2_ 120–133, *c*_3_–*c*_3_ 272–280, *d*_1_–*d*_1_ 85–100, *d*_2_–*d*_2_ 233–248, *e*_1_–*e*_1_ 76–90, *e*_2_–*e*_2_ 154–171, *f*_2_–*f*_2_ 104–130, *h*_1_–*h*_1_ 38–64, *c*_1_–*d*_1_ 59–76, *d*_1_–*e*_1_ 79–91, *e*_1_–*f*_2_ 59–86, *f*_2_–*h*_1_ 25–28. Hysterosoma dorsally with a pattern may be resulting in highly wrinkled ornamentation that somewhat round convex on protuberances, oblique wide ridges laterally, and opisthosoma dorsally with longitudinal ridges.

Venter (Figure 9 and Figure 10A). Striae mostly transverse, pregenital striae transverse and broken. Genital flap with transverse striae. All ventral setae thin and smooth. Setae *1a* as long as distance between their bases; setae *3a* and *4a* shorter than distance between their bases. Coxal setae count 2-1-1-1, one pair of pseudanal setae (*ps*_1_), two pairs of smooth genital setae (*g*_1–2_). Length of setae: *1a* 19–22, *3a* 18–23, *4a* 14–21, *ag* 13–17, *g*_1_ 25–36, *g*_2_ 25–35, *ps*_1_ 11–15. Distances between setal bases: *1a*–*1a* 19–28, *3a*–*3a* 37–46, *4a*–*4a* 80–92, *ag*–*ag* 20–25, *g*_1_–*g*_1_ 25–32, *g*_2_–*g*_2_ 48–63, *ps*_1_–*ps*_1_ 16–44.

Gnathosoma (Figure 10B,C). Stylophore with well-developed bilobed horn-like anterior projections. Integument of base of stylophore with longitudinal striae, projections with granulate pattern (Figure 10B). Ventral infracapitular setae *m* smooth, 18–19 in length. Palp setation and notation as shown in Figure 10C. Dorsal surface of palp base with a pair of inconspicuous supracoxal setae (*e*). Palptarsus: terminal eupathidium (*suζ*) club-like with blunt tip end, 4.4–5.9 long, two lateral eupathidia, *ul′ζ* 4.4–5.8 and *ul″ζ* 4.3–5.3 long, one solenidion (*ω*) 2.9–4 long; three short, smooth, tactile setae (*a*, *b*, *c*).

Legs (Figure 11A–D). Empodial claws absent. One pair of duplex setae on tarsus I, solenidion *ω′* 8–9, one additional ventral solenidion (*vω*) at the same transverse level with *u* setae, 10–14 long, tectal seta (*tc′*) unpaired, thicker than other tactile setae on tarsus I; tibia I with one solenidion *φ* 10–12 long; tarsus II without duplex setae, solenidion *ω″* 10–13 long, tectal seta (*tc′*) unpaired, thicker than other tactile setae; tarsus III without solenidion, tectal setae paired; tarsus IV with one proximal solenidion *ω′* 6–7 long, tectal setae paired. Number of tactile setae on leg (I–IV) segments: trochanters 1-1-1-1, femora 6-5-3-1, genua 1-1-1-1, tibiae 3-1-1-1, tarsus 7-7-6 (or 7) -6 (or 7). Number of eupathidia on tarsus I–IV: 3-3-0-0. Legs I–IV setation and notation as shown in Figure 11A–D. Length of leg segments: femur I 99–110, genu I 41–46, tibia I 39–47, tarsus I 67–74; femur II 81–90, genu II 38–45, tibia II 31–38, tarsus II 62–71; femur III 67–78, genu III 33–36, tibia III 37–41, tarsus III 75–78; femur IV 96–102, genu IV 33–44, tibia IV 44–45, tarsus IV 79–85.

Number of tactile setae on Tarsus III–IV varies among specimens, and differs between right and left legs in the same specimen (Table 2). Among 23 adult females, 12 (one from Thailand, one from India, six from Yunnan Province, China and four from Guangdong Province, China) with six tactile setae (*u′*, *u″*, *ft′*, *ft″*, *pv′*, *pv″*) on both right and left tarsus III; six (two from Thailand, four from Yunnan Province, China) with seven tactile setae (*u′*, *u″*, *ft′*, *ft″*, *pv′*, *pv″*, *v′*_1_) on both right and left tarsus III; three (two from Yunnan Province, China and one from Guangdong Province, China) with seven tactile setae on right tarsus III and six tactile setae on left; one (from Guangdong Province China)with six tactile setae (*u′*, *u″*, *ft′*, *ft″*, *pv′*, *pv″*) and one solenidion (*ω′*) on right tarsus III and six tactile setae on left; one (from Yunnan Province, China) with seven tactile setae on left tarsus III and unknown the right side due to the broken tarsus III. Among 23 adult females, 15 (one from Thailand, one from India, eight from Yunnan Province, China and five from Guangdong Province, China) with six tactile setae (*u′*, *u″*, *ft′*, *ft″*, *pv′*, *pv″*) and one solenidion (*ω′*) on both right and left tarsus IV; four (two from Thailand, two from Yunnan Province, China) with seven tactile setae (*u′*, *u″*, *ft′*, *ft″*, *pv′*, *pv″*, *v′*_1_) and one solenidion (*ω′*) on both right and left tarsus IV; two (from Yunnan Province, China) with seven tactile setae and one solenidion on left tarsus IV and six tactile setae and one solenidion on right; one (from Guangdong Province, China) with seven tactile setae on left tarsus IV and six tactile setae and one solenidion on right; one (from Yunnan Province, China) with six tactile setae and one solenidion on left tarsus IV and unknown the right side due to the broken tarsus IV. The variations in the setal count of tarsus III and IV are here considered intraspecific in nature, and attributed to the geographical position of the samples and different host plant species. In order to express the ontogenetic development of leg chaetotaxy conveniently, tarsus III with six tactile setae and tarsus IV with six tactile setae and one solenidion are regarded as normal setal count.

Setal counts (solenidion in parentheses following tactile setae) on legs I–IV are: femora 6-5-3-1, genua 1-1-1-1, tibiae 3(1)-1-1-1, tarsus 7(2)-7(1)-6(0)-6(1). There is a significant amount of setal suppression on the legs in this species, with a total of 15 setae being added to the legs in the adult female stage of this species: pair *l*_1_ on femur I, pair *v*_1_ on tarsus I, pair *l*_1_ on femur II, pair *v*_1_ on tarsus II, *v′* on trochanter III, *v′* and *l′*_1_ on femur III, *l′* on genua III, *v′* on trochanter IV, *l″* on genua IV and *ω′* on tarsus IV. According to the normal ontogenetic setal additions for the family [10], seven of thirteen additional setae are delayed additions: *v′*_1_ on tarsus I suppressed on protonymph stage, *v″*_1_ on tarusus I, *v′*_1_ on tarsus II, *v′* on trochanter III and IV, *l* on genua III and IV are suppressed on deutonymph stage. 

Male (*n* = 9)

Dorsum (Figure 12). Idiosoma gradual narrowing caudally, 184–209 long, 123–147 wide, with length much longer than width. Dorsum without a protuberance.

Prodorsum with three pairs of palmate setae, covered with short longitudinally aligned spinules, *v*_2_ 19–21, *sc*_1_ 32–37 and *sc*_2_ 19–21. Distances between setal bases: *v*_2_–*v*_2_ 50–54, *sc*_1_–*sc*_1_ 74–76, *sc*_2_–*sc*_2_ 133–145. Integument with irregular fine granulate.

Hysterosoma with 11 pairs of setae (*c*_1–3_, *d*_1–2_, *e*_1–2_, *f_2_*, *h*_1–3_), similar in shape to prodorsal setae, except with setae *h*_2–3_ of differing morphology, similar to other ventral setae and inserted posterodorsally. Seta *c*_2_ slightly shorter than other dorsal setae. Dorsal central setae (*c*_1_, *d*_1_, *e*_1_) much shorter than the distance to setae in the next setal row. Length of setae: *c*_1_ 17–19, *c*_2_ 13–15, *c*_3_ 24–26, *d*_1_ 16–18, *d*_2_ 19–20, *e*_1_ 13–16, *e*_2_ 20–22, *f*_2_ 21–26, *h*_1_ 17–23, *h*_2_ 13–23, *h*_3_ 13–16. Distances between setal bases: *c*_1_–*c*_1_ 21–25, *d*_1_–*d*_1_ 37–40, *e*_1_–*e*_1_ 25–25, *f*_2_–*f*_2_ 57–59, *h*_1_–*h*_1_ 28–31. Hysterosoma dorsally with irregular fine granulate, except for band of transverse striae between paired *c*_1_ and *d*_1_. 

Venter (Figure 13). Striae mostly transverse. All ventral setae thin and smooth. Coxal setae count 2-1-1-1. Length of setae: *1a* 20–21, *3a* 16–21, *4a* 13–15, *ag* 14–17, *g*_1_ 6–8, *g*_2_ 5–9, *ps*_1_ 5–6. Distances between setal bases: *1a*–*1a* 22–24, *3a*–*3a* 31–38, *4a*–*4a* 34–38, *ag*–*ag* 7–8.

Gnathosoma (Figure 10B–D). Stylophore with short bilobed horn-like anterior projections as shown in Figure 10B. Ornamentation of integument similar to that of female. Ventral infracapitular setae *m* smooth, 11–17 in length. Palp setation and notation as shown in Figure 10D. Palptarsus: terminal eupathidium (*suζ*) club-like with sharp tip end, 2.8–3 long, two lateral eupathidia, *ul′ζ* 5–6 and *ul″ζ* 5–7 long, one solenidion (*ω*) 3–5 long; three short, smooth, tactile setae (*a*, *b*, *c*). 

Legs (Figure 14A–D). Empodial claws absent. One pair of duplex setae on tarsus I, solenidion *ω′* 12–14, *ω″*_1_ 8–11 long, one additional ventral solenidion (*vω*) at the same level with *u* setae, 13–16 long, tectal setae (*tc′*) unpaired, thicker than other tactile setae on tarsus I; tibia I with four solenidion, *φ* 12–13, *φ*′ 8–9, *φ″* 6–8, *φ″*_1_ 8–10 long; tarsus II without duplex setae, solenidion *ω″* 11–14 long, tectal setae (*tc′*) unpaired, thicker than other tactile setae; tibia II with two solenidion, *φ′* 8–9, *φ″* 8–9 long, tarsus III with one solenidion *ω′* 7–9, *tc* paired; tarsus IV with one proximal solenidion *ω′* 7–9 long, *tc* paired. Number of tactile setae on leg (I–IV) segments: trochanters 1-1-1-1, femora 6-5-3-1, genua 2-2-1-1, tibiae 3-1-1-1, tarsus 7-7-7-7. Number of eupathidia on tarsus I–V: 3-3-0-0. Legs I–IV setation and notation as shown in Figure 14A–D. Length of leg segments: femur I 78–86, genu I 40–42, tibia I 46–47, tarsus I 67–68; femur II 74–76, genu II 39–42, tibia II 39–39, tarsus II 64–64; femur III 78–79, genu III 33–37, tibia III 32–40, tarsus III 69–73; femur IV 81–85, genu IV 35–38, tibia IV 44–44, tarsus IV 75–80.

The number of tactile setae on tarsus III and tarsus IV varies among specimens, and differs between right and left legs in the same specimen. Among nine adult males, seven (three from Thailand, four from Yunnan Province, China) with seven tactile setae (*u′*, *u″*, *ft′*, *ft″*, *pv′*, *pv″*, *v′*_1_) and one solenidion (*ω′*) on both right and left tarsus III; one (from Yunnan Province, China) with eight tactile setae (*u′*, *u″*, *ft′*, *ft″*, *pv′*, *pv″*, *v′*_1_, *v″*_1_) and one solenidion (*ω′*) on both right and left tarsus III; one (from Yunnan Province, China) with eight tactile setae and one solenidion on left and seven tactile and one solenidion on right. Among nine adult males, four (from Yunnan Province, China) with six tactile setae (*u′*, *u″*, *ft′*, *ft″*, *pv′*, *pv″*) and one solenidion (*ω′*) on both right and left tarsus IV; three (two from Thailand, one from Yunnan Province, China) with 7 tactile setae (*u′*, *u″*, *ft′*, *ft″*, *pv′*, *pv″*, *v′*_1_) and one solenidion (*ω′*) on both right and left tarsus IV; and two (one from Thailand, one from Yunnan Province, China) with eight tactile (*u′*, *u″*, *ft′*, *ft″*, *pv′*, *pv″*, *v′*_1_, *v″*_1_) and one solenidion (*ω′*) on right Tarsus IV and seven tactile setae and one solenidion on left. The variations in the setal count of tarsus III and IV are here considered intraspecific in nature, and attributed to the geographical position of the samples and different host plant species. In order to express the ontogenetic development of leg chaetotaxy conveniently, tarsus III–IV with seven tactile setae and one solenidion are regarded as normal setal counts. 

The male has a slightly different chaetotactic formula to the female. Setal counts on legs I–IV: femur 6-5-3-1, genua 2-2-1-1, tibia 3(4)-1(2)-1-1, tarsus 7(3)-7(1)-7(1)-7(1). The male adds four more tactile setae to legs than the dose for the female, accurately, *l’* on genua III and IV, *v″*_1_ on tarsus III and IV. The male also adds more solenidia to the legs than does the female: *φ’*, *φ″*, *φ″*_1_ on tibia I, *ω″*_1_ on tarsus I, *φ’*, *φ″* on tibia II and *ω′* on tarsus III.

A total of 26 setae were added to the legs in the adult male stage of this species, and 11 additional setae are delayed additions: *v′* on Genua I and II suppressed in larva stage, *v′*_1_ on tarsus I suppressed on protonymph stage, and *v″*_1_, *ω″*_1_ on tarusus I, *v′*_1_ on tarsus II, *v′* on trochanter III and IV, *l′* on genua III and IV, and *ω′* on tarsus III are suppressed on deutonymph stage. Tarsus III in male of *S. vannus* does not express standard adult seta *v′*_1_, replaced by *v″*_1_.

Aedeagus (Figure 13A,B). Aedeagus dorsally curved, gradually narrowing and bent distally to form a somewhat right angle. 

Deutonymph (*n* = 6)

Dorsum (Figure 15). Idiosoma oval without protuberance on hysterosoma. 190–268 long, 168–230 wide.

Prodorsum with three pairs of club-like setae, covered with short longitudinally aligned spinules, *v*_2_ 25–26, *sc*_1_ 30–31 and *sc*_2_ 26–30. Distances between setal bases: *v*_2_–*v*_2_ 45–46, *sc*_1_–*sc*_1_ 72–78 and *sc*_2_–*sc*_2_ 108–173. Integument with irregular fine granulate medially and longitudinal stiae laterally. 

Hysterosoma with 11 pairs of setae (*c*_1–3_, *d*_1–2_, *e*_1–2_, *f*_2_, *h*_1–3_), similar in shape to prodorsal setae, except with setae *h*_2–3_ of differing morphology, similar to other ventral setae and inserted posteroventrally. Seta *c*_2_ much shorter than other dorsal setae. Dorsal central setae (*c*_1_, *d*_1_, *e*_1_) shorter than the distance to setae in the next setal row. Length of setae: *c*_1_ 27–30, *c*_2_ 13–15, *c*_3_ 24–26, *d*_1_ 62–68, *d*_2_ 25–25, *e*_1_ 24–27, *e*_2_ 26–29, *f*_2_ 25–26, *h*_1_ 22–25, *h*_2_ 12–13, *h*_3_ 14–15. Distances between setal bases: *c*_1_–*c*_1_ 26–32, *d*_1_–*d*_1_ 40–42, *e*_1_–*e*_1_ 26–34, *f*_2_–*f*_2_ 51–61, *h*_1_–h_1_ 33–34. Hysterosoma dorsally with irregular fine granulate, except for band of transverse striae between paired *c*_1_ and *d*_1_, and oblique broken striae on opisthosoma.

Venter. Striae mostly transverse. All ventral setae thin and smooth. Setae *1a* as long as distance between their bases; setae *3a* and *4a* shorter than distance between their bases. Coxal setae count 2-1-1-1, one pair of pseudanal setae (*ps*_1_) and one pair of smooth genital setae (*g*_1_). Length of setae: *1a* 12–15, *3a* 11–16, *4a* 9–10, *ag* 9–11, *g*_1_ 9–10, *ps*_1_ 7–7. Distances between setal bases: *1a*–*1a* 26–31, *3a*–*3a* 40–47, *4a*–*4a* 54–58, *ag*–*ag* 10–10.

Gnathosoma (Figure 10F and Figure 15). Stylophore with slightly or well-developed bilobed horn-like anterior projections. Ornamentation of integument similar to that of female. Ventral infracapitular setae *m* smooth, 13–16 in length. Length of setae on palptarsus: *suζ* 3–3.4, *ul′ζ* 4.8–6, *ul″ζ* 5.5–7, *ω* 2–2.8.

Legs (Figure 16A–D). Empodial claws absent. One pair of duplex setae on tarsus I, sometimes setal bases of *ft′* and *ω′* separated, solenidion *ω′* 5–6, one additional ventral solenidion (*vω*) at the same level with *u* setae, 7–10 long, tectal seta (*tc′*) unpaired, thicker than other tactile setae on tarsus I; tibia I with one solenidion 5–8 long; tarsus II without duplex setae, solenidion *ω″* 8–10 long, tectal seta (*tc′*) unpaired, thicker than other tactile setae; and tarsus III and tarsus IV without solenidion. Number of tactile setae on leg (I–IV) segments: trochanters 1-1-1-0, femora 4-3-1-1, genua 1-1-0-0, tibiae 3-1-1-1, tarsus 5-5-6-6. Number of eupathidia on tarsus I–V: 3-3-0-0. Legs I–IV setation and notation as shown in Figure 16A–D. Length of leg segments: femur I 46–53, genu I 18–22, tibia I 21–26, tarsus I 36–39; femur II 40–44, genu II 18–19, tibia II 18–20, tarsus II 32–37; femur III 34–40, genu III 15–19, tibia III 17–22, tarsus III 35–38; femur IV 34–39, genu IV 15–18, tibia IV 18–22, tarsus IV 30–41.

Four setae are added to the legs of the deutonymph of this species during ontogeny: *l′* is added to femur I. *v′* on trochanter I–III, respectively. A total of 50 setae are suppressed on legs I–IV in the deutonymphal stage of this species: two on femur I, four on genua I, four on tibia I, eight on tarsus I, four on genua II, four on tibia II, three on tarsus II, one on femur III, three on genua III, four on tibia III, three on tarsus III, one on femur IV, three on genua IV, four on tibia IV, and two on tarsus IV.

Protonymph (*n* = 1)

Dorsum (Figure 17 and Figure 18). Idiosoma oval without protuberance on hysterosoma, 198 long, 148 wide.

Prodorsum with three pairs of club-like setae, covered with short longitudinally aligned spinules, *v*_2_ 50, *sc*_1_ 44 and *sc*_2_ 32. Distances between setal bases: *v*_2_–*v*_2_ 50, *sc*_1_–*sc*_1_ 82, *sc*_2_–*sc*_2_ 148. Integument with irregular fine granulate medially and broken longitudinal stiae laterally. 

Hysterosoma with 11 pairs of setae (*c*_1–3_, *d*_1–2_, *e*_1–2_, *f*_2_, *h*_1–3_), similar in shape to prodorsal setae, except with setae *h*_2–3_ is of differing morphology, similar to other ventral setae and inserted posteroventrally. Seta *c*_2_ much shorter than other dorsal setae. Dorsal central setae (*c*_1_, *d*_1_, *e*_1_) slightly longer than the distance to setae in the next setal row. Length of setae: *c*_1_ 43, *c*_2_ 22, *c*_3_ 34, *d*_1_ 42, *d*_2_ 44, *e*_1_ 37, *e*_2_ 38, *f*_2_ 30, *h*_1_ 29, *h*_2_ 12, *h*_3_ 10. Distances between setal bases: *c*_1_–*c*_1_ 29, *d*_1_–*d*_1_ 48, *e*_1_–*e*_1_ 35, *f*_2_–*f*_2_ 63, *h*_1_–*h*_1_ 36. Hysterosoma dorsally with irregular fine granulate, except for band of transverse striae between paired *c*_1_ and *d*_1_, narrow band of transverse striae between *d*_1_ and *e*_1_, and oblique broken striae on opisthosoma.

Venter. Striae mostly transverse. All ventral setae thin and smooth. Setae *1a* and *3a* shorter distance between their bases. Coxal setae count 2-1-1-0, one pair of pseudanal setae (*ps*_1_). Length of setae: *1a* 13, *3a* 33, *ag* 11, *ps*_1_ 7. Distances between setal bases: *1a*–*1a* 24, *3a*–*3a* 33, *ag*–*ag* 11.

Gnathosoma (Figure 17). Stylophore with slightly bilobed horn-like anterior projections. Ornamentation of integument similar to that of female and deutonymph. Ventral infracapitular setae *m* smooth, 12 in length. Length of setae on palptarsus: *suζ* 3.6, *ul′ζ* 5.6, *ul″ζ* 6.7, *ω* 2.

Legs (Figure 19A–D). Empodial claws absent. One pair of duplex setae on tarsus I, sometimes setal bases of *ft′* and *ω′* separated, solenidion *ω′* 5, one additional ventral solenidion (*vω*) at the same level with *u* setae, 8 long, tectal seta (*tc′*) unpaired, thicker than other tactile setae on tarsus I; tibia I with one solenidion *φ* 7 long; tarsus II without duplex setae, solenidion *ω″* 6 long, tectal seta (*tc′*), thicker than other tactile setae; tarsus III and tarsus IV without solenidion. Number of tactile setae on leg (I–IV) segments: trochanters 0-0-0-0, femora 3-3-1-1, genua 1-1-0-0, tibiae 3-1-1-1, tarsus 5-5-6-6. Number of eupathidia on tarsus I–V: 3-3-0-0. Legs I–IV setation and notation as shown in Figure 19A–D. Length of leg segments: femur I 32, genu I 17, tibia I 16, tarsus I 28; femur II 26, genu II 15, tibia II 15, tarsus II 28; femur III 25, genu III 13, tibia III 15, tarsus III 32; femur IV 22, genu IV 11, tibia IV 12, tarsus IV 29.

As we do not have the larva to examine, we cannot determine which setae are added to the legs in the protonymph, although based on what is already known for the ontogenetic setal additions for the family, it would appear that the protonymph maintains the larval chaetotaxy on femora I–IV, genua I–IV, tibiae I–IV and tarsus III–IV, adding only the tectal (*tc″*) to tarsus I–II and *ω′* to tarsus I, as is normal for the family. A total of 34 setae are suppressed on legs I–IV in the larval-protonymphal stage of this species: three on genua I, two on tibia I, four on tarsus I, three on genua II, four on tibia II, two on tarsus II, one on femur III, two on genua III, four on tibia III, two on tarsus III, one on femur IV, two on genua IV, and four on tibia IV.

### 3.3. New Species

Family Tetranychidae Donnadieu

Subfamily Tetranychinae Berlese

Tribe Aponychini Rimando & Corpuz-Raros

Genus *Stylophoronychus* Prasad


*Stylophoronychus wangae*
**Pan, Jin & Yi sp. nov.**


**Material examined.** Holotype, one female, ex. bamboo, from Majiang Country, Guizhou Province, China, on 3 August 2020, coll. Tian-Ci Yi. Paratype, 14 females, three males, one deutonymph, the same data as the holotype. All deposited at the Institute of Entomology, Guizhou University, Guiyang, P.R. China (GUGC).

**Etymology.** The name of the new species is named after the late Professor Huifu Wang in honor of her contributions to Acarology in China.


**Description**


Figure 20, Figure 21, Figure 22, Figure 23, Figure 24, Figure 25, Figure 26, Figure 27, Figure 28, Figure 29, Figure 30, Figure 31, Figure 32, Figure 33, Figure 34, Figure 35 and Figure 36

Female (*n* = 15)

Dorsum (Figure 20, Figure 21 and Figure 22). Body oblong, 359 (311–366) long excluding gnathosoma, 495 (422–506) including gnathosoma, 277 (251–277) wide. Color: brownish yellow, with some black patches on the dorsum and two pairs of red eyes. Dorsocentral region idiosoma with a distinct convex protuberance. Integument finely granulate and with irregular striae. Prodorsum with three pairs of setae (*v*_2_, *sc*_1_, *sc*_2_), seta *v*_2_ spatulate, on two slightly developed anterior lobes, two times as long as wide, more than twice as long as *sc*_1_, seta *sc*_1_ smaller, fan-shaped; setae *sc*_2_ set on strong tubercles, linear, all covered with short barbs; *v*_2_ 26 (25–30), *sc*_1_ 14 (12–15), *sc*_2_ 59 (50–62), *v*_2_–*v*_2_ 87 (81–90), *sc*_1_–*sc*_1_ 129 (120–129), *sc*_2_–*sc*_2_ 244 (222–244).

Hysterosomal dorsum with convex bulge that bears setae *c*_1_, *d*_1_ and *e*_1_, oblique wide ridges laterally, full of irregular wrinkles and finely granulated; the dorsocentral setae (*c*_1_, *d*_1_, *e*_1_) long linear, similar in shape to *sc*_2_, the dorsolateral setae (*c*_2_, *d*_2_, *e*_2_) are greatly different in morphology and size, setae *c*_2_ and *d*_2_ spatulate but the former smaller, seta *e*_2_ long linear; setae *e*_2_, *f*_2_, *h*_1_ are nearly the same length and similar in shape to the dorsocentral setae. The length of dorsal central setae (*c*_1_, *d*_1_, *e*_1_) is equal to or longer than the distances between the seta and the next setal row (*c*_1_–*d*_1_, *d*_1_–*e*_1_). Length of dorsal setae: *c*_1_ 68 (53–68), *c*_2_ 13 (11–13), *c*_3_ 62 (54–62), *d*_1_ 66 (56–72), *d*_2_ 19 (15–23), *e*_1_ 56 (45–57), *e*_2_ 61 (52–61), *f*_2_ 57 (53–62), *h*_1_ 58 (48–58); distance between dorsal setae: *c*_1_–*c*_1_ 39 (39–47), *c*_2_–*c*_2_ 168 (151–175), *c*_3_–*c*_3_ 277 (251–277), *d*_1_–*d*_1_ 54 (41–54), *d*_2_–*d*_2_ 221 (206–230), *e*_1_–*e*_1_ 42 (35–44), *e*_2_–*e*_2_ 165 (150–165), *f*_2_–*f*_2_ 115 (99–115), *h*_1_–*h*_1_ 60 (53–60), *c*_1_–*d*_1_ 53 (39–53), *d*_1_–*e*_1_ 61 (41–67), *e*_2_–*f*_2_ 118 (103–120), *e*_1_–*h*_1_ 40 (33–40).

Venter (Figure 23 and Figure 24). Striae mostly transverse, pregenital striae with discontinuous slight fine lines. Genital flap with transverse striae, oblique striae anterior-laterally, longitudinal medially and transverse striae posteriorly. All ventral setae thin and smooth. Setae *1a*, *3a* and *4a* shorter than distance between their bases, respectively. Coxal setal count 2-1-1-1, one pair of anal setae (*ps*_1_), two pairs of genital setae (*g*_1–2_). Length of ventral setae: *1a* 18 (16–20), *3a* 22 (12–23), *4a* 15 (15–19), *1b* 25 (23–28), *1c* 28 (26–31), *2b* 22 (17–28), *3b* 17 (17–28), *4b* 26 (22–26); distance between intercoxal and coxae setae: *1a*–*1a* 22 (21–27), *3a*–*3a* 59 (55–67), *4a*–*4a* 82 (53–82); aggenital setae: *ag* 13 (9–15), *ag*–*ag* 17 (17–21); genital setae: *g*_1_ 31 (29–31), *g*_2_ 40 (34–40), *g*_1_–*g*_1_ 25 (24–28), *g*_2_–*g*_2_ 55 (49–55); anal setae one pair: *ps*_1_ 14 (10–14), *ps*_1_–*ps*_1_ 28 (18–28); para-anal setae two pairs *h*_2_ 31 (25–32), *h*_3_ 32 (27–32), *h*_2_–*h*_2_ 57 (28–57), *h*_3_–*h*_3_ 77 (48–77).

Gnathosoma (Figure 25 and Figure 30C). Stylophore with longitudinal striae, having two strong lobes distally. Ventral infracapitular setae *m* smooth, 19 (14–19) in length. *m*–*m* 34 (31–34). Palp setation and notation as shown in Figure 25. Palptarsus: terminal eupathidium (*suζ*) elongate, blunt tipped, 4.4 (3.5–5.8) in length, 2.4 (2.2–2.9) in width; two lateral eupathidia (*ul′ζ* and *ul″ζ*) subequal in length, 4.8 (4.2–5.8); one solenidion (*ω*), 3.1 (2.2–3.1); three tactile setae: *a* 4.5 (4.5–6.3), *b* 5.3 (5.3–6.5), *c* 7.5 (4.9–7.6). Measurements of setae on other palp segments: *dPFe* 36 (32–38), *l″PGe* 17 (17–19), *dPTi* 11 (7–11), *l′PTi* 12 (9–12), *l″PTi* 14 (14–18). Peritreme slightly enlarged at distal end (Figure 30C).

Legs (Figure 26). Tarsus I with one pair of duplex setae and one additional ventral solenidion (*vω*) at the same transverse level with *u*. Two solenidia *vω* 12 (11–14), *ω′* 13 (9–13), single *tc* on tarsus I (*tc″* absent); tibia I with one solenidion *φ* 13 (12–14) long; tarsus II with one solenidion *ω″* 12 (11–12) long; tarsi III and IV with one solenidion *ω′* 8 (7–10), *ω′* 6 (6–10), respectively. Segmental length of legs: leg I: trochanter 31 (27–32), femur 95 (93–102), genua 51 (45–51), tibia 58 (46–58), tarsus 80 (65–84); leg II: trochanter 26 (20–27), femur 79 (75–80), genu 44 (38–47), tibia 43 (35–43), tarsus 70 (57–76); leg III: trochanter 26 (20–26), femur 72 (62–72), genua 38 (33–38), tibia 50 (41–50), tarsus 80 (68–81); leg IV: trochanter 31 (24–31), femur 98 (84–98), genua 44 (38–44), tibia 56 (50–56), tarsus 92 (79–92); legs chaetotaxy I–IV (eupathidia and solenidia in parentheses): trochanters 1-1-1-1, femora 6-5-3-1, genua 2-1-1-1, tibiae 3(0)(1)-2-1-1, tarsi 6(3)(2)-6(3)(1)-6(0)(1)-6(0)(1).

Male (*n* = 3)

Dorsum (Figure 27 and Figure 28). Idiosoma subovate, narrowing posteriorly, brownish yellow, with some black pathes on the dorsum and two pairs of red eyes. Length of idiosoma 208 (208–217) long excluding gnathosoma, 271 (271–274) including gnathosoma, 183 (183–189) wide. Hysterosoma dorsally with irregular fine granulate, except for band of transverse striae between paired *c*_1_, *d*_1_ and *e*_1_. The 13 pairs of dorsal setae shorter than those of female, mostly spatulate. Length of dorsal setae: *v*_2_ 17 (17–20), *sc*_1_ 10 (10–11), *sc*_2_ 18 (17–18), *c*_1_ 16 (16–20), *c*_2_ 7, *c*_3_ 27 (27–30), *d*_1_ 15 (15–17), *d*_2_ 13 (11–13), *e*_1_ 17 (14–17), *e*_2_ 20 (20–29), *f*_2_ 28 (28–29), *h*_1_ 26; distance between dorsal setae: *v*_2_–*v*_2_ 60 (59–60), *sc*_1_–*sc*_1_ 87, *sc*_2_–*sc*_2_ 178 (177–178), *c*_1_–*c*_1_ 21 (19–21), *c*_2_–*c*_2_ 99 (99–102), *c*_3_–*c*_3_ 173 (162–173), *d*_1_–*d*_1_ 33 (29–33), *d*_2_–*d*_2_ 117 (110–117), *e*_1_–*e*_1_ 19 (18–19), *e*_2_–*e*_2_ 83 (83–86), *f*_2_–*f*_2_ 64, *h*_1_–*h*_1_ 37 (37–38), *c*_1_–*d*_1_ 26 (26–31), *d*_1_–*e*_1_ 31 (29–31), *e*_1_–*f*_2_ 59 (56–59), *f*_2_–*h*_1_ 19 (17–19).

Venter (Figure 29). Striae mostly transverse. All ventral setae thin and smooth. Setae *1a*, *3a* and *4a* shorter than distance between their bases respectively. Coxal setal count 2-1-1-1, one pair of anal setae (*ps*_1_), two pairs of genital setae (*g*_1–2_). Length of ventral setae: *1a* 20, *3a* 20, *4a* 14 (14–17), *1b* 19 (19–23), *1c* 17 (17–22), *2b* 16 (16–23), *3b* 20 (20–24), *4b* 20 (18–20); distance between intercoxal and coxae setae: *1a*–*1a* 22 (19–22), *3a*–*3a* 43 (38–43), *4a*–*4a* 32 (32–37); aggenital setae: *ag* 16 (15–16), *ag*–*ag* 5 (5–6); genital setae: *g*_1_ 5 (5–7), *g*_2_ 8, *g*_1_–*g*_1_ 17 (13–17), *g*_2_–*g*_2_ 29 (25–29); anal setae one pair: *ps*_1_ 8 (8–9), *ps*_1_–*ps*_1_ 20; para-anal setae two pairs *h*_2_ 12 (6–12), *h*_3_ 12 (12–13), *h*_2_–*h*_2_ 15 (15–18), *h*_3_–*h*_3_ 35 (35–37).

Gnathosoma (Figure 27, Figure 30A,B and Figure 31). Stylophore with short bilobed horn-like anterior projections as shown in Figure 27. Subcapitular setae *m* smooth, 16 (13–16) in length, *m*–*m* 28 (27–28). Palp setation and notation as shown in Figure 31. Palptarsus: terminal eupathidium (*suζ*) elongate, blunt tipped, 3.4 (3.4–3.8) in length, 1.6 (1.4–1.6) in width; two lateral eupathidia (*ul′ζ* and *ul″ζ*) subequal in length, 3.9 (3.8–3.9); one solenidion (*ω*), 2.4 (2.4–2.5); three tactile setae: *a* 3.8 (3.8–5.3), *b* 3.8 (3.8–6.5), *c* 3.4. Measurements of setae on other palp segments: *dPFe* 17 (15–17), *l″PGe* 9 (9–11), *dPTi* 6.5 (6.5–10.1), *l′PTi* 8.9 (8.9–9), *l″PTi* 7.9 (6.9–10.8). Peritreme ending in small expansion (Figure 30A,B).

Aedeagus (Figure 30E,F). Aedeagus dorsally curved, gradually narrowing and distally dipping upturned forming an acute angle, blunt tipped. 

Legs (Figure 32). Tarsus I with one pair of duplex setae, one additional ventral solenidion (*vω*) and one additional dorsal solenidion *ω″*_1_, three solenidia, *ω′* 14 (13–14), *vω* 11 (11–13) long, *ω″*_1_ 13 (13–14) long; tibia I with three solenidia, *φ* 14 (9–14), *φ′* 11 (11–13), *φ″* 14 (14–16) long; tarsus II with two solenidia *ω″* 16 (13–16), *ω″*_1_ 12 long; tibia II with one solenidion, *φ* 10 (10–12); tarsi III and IV with one solenidion *ω′* 12 (11–12), *ω′* 10 (9–10), respectively. Segmental length of legs: leg I: trochanter 27, femur 86 (86–94), genua 53 (53–54), tibia 58 (58–60), tarsus 67 (67–72); leg II: trochanter 22, femur 70 (70–77), genua 46 (46–47), tibia 48 (47–48), tarsus 59 (59–68); leg III: trochanter 20 (20–22), femur 63 (63–65), genua 33 (33–37), tibia 49 (49–50), tarsus 72 (72–74); leg IV: trochanter 26 (22–26), femur 81 (81–87), genua 41, tibia 55 (55–56), tarsus 72 (72–80); legs chaetotaxy I–IV (eupathidia and solenidia in parentheses): trochanters 1-1-1-1, femora 7-5-3-1, genua 3-3-1-1, tibiae 3(0)(3)-2(0)(1)-1-1, tarsi 6(3)(3)-6(3)(2)-6(0)(1)-6(0)(1).

Deutonymph (*n* = 1)

Dorsum (Figure 33). Length of idiosoma 243 long excluding gnathosoma, 300 including gnathosoma, 211 wide. Integument finely granulated, having irregular wrinkles, slightly uplifted in the middle. The shape of dorsal setae similar to female and the length of dorsal central setae (*c*_1_, *d*_1_, *e*_1_) is much longer than the distances between bases of setae and setae in next row (*c*_1_–*d*_1_, *d*_1_–*e*_1_). Length of dorsal setae: *v*_2_ 48, *sc*_1_ 13, *sc*_2_ 43, *c*_1_ 55, *c*_2_ 9, *c*_3_ 43, *d*_1_ 58, *d*_2_ 18, *e*_1_ 42, *e*_2_ 46, *f*_2_ 47, *h*_1_ 44; distance between dorsal setae: *v*_2_–*v*_2_ 66, *sc*_1_–*sc*_1_ 107, *sc*_2_–*sc*_2_ 205, *c*_1_–*c*_1_ 40, *c*_2_–*c*_2_ 141, *c*_3_–*c*_3_ 211, *d*_1_–*d*_1_ 42, *d*_2_–*d*_2_ 180, *e*_1_–*e*_1_ 31, *e*_2_–*e*_2_ 119, *f*_2_–*f*_2_ 82, *h*_1_–*h*_1_ 44, *c*_1_–*d*_1_ 28, *d*_1_–*e*_1_ 34, *e*_2_–*f*_2_ 21, *e*_1_–*h*_1_ 90.

Venter (Figure 34). Ventral striae mostly transverse except for pregenital area with longitudinal striae, oblique striae anterior-laterally, longitudinal medially and transverse striae posteriorly. All ventral setae thin and smooth. Setae *1a*, *3a* and *4a* shorter than distance between their bases, respectively. Coxal setal count 2-1-1-1, one pair of anal setae (*ps*_1_), two pairs of genital setae (*g*_1–2_). Length of ventral setae: *1a* 11, *3a* 14, *4a* 13, *1b* 20, *1c* 18, *2b* 17, *3b* 13, *4b* 12; distance between intercoxal and coxae setae: *1a*–*1a* 25, *3a*–*3a* 45, *4a*–*4a* 65; aggenital setae: *ag* 10, *ag*–*ag* 13; genital setae: *g*_1_ 19, *g*_1_–*g*_1_ 31; anal setae one pair: *ps*_1_ 7, *ps*_1_–*ps*_1_ 11; para-anal setae two pairs *h*_2_ 14, *h*_3_ 19, *h*_2_–*h*_2_ 19, *h*_3_–*h*_3_ 34.

Gnathosoma (Figure 30D, Figure 33 and Figure 35). Stylophore with two well-developed lobes distally as shown in Figure 33. Subcapitular setae *m* smooth, 13 in length, *m*–*m* 30. Palp setation and notation as shown in Figure 35. Palptarsus: terminal eupathidium (*suζ*) elongate, blunt tipped, 5.3 in length, 1.5 in width; two lateral eupathidia (*ul′ζ* and *ul″ζ*) subequal in length, 3.4; one solenidion (*ω*), 2.3; three tactile setae: *a* 4.3, *b* 3.2, *c* 3. Measurements of setae on other palp segments: *dPFe* 32, *l″PGe* 16, *dPTi* 7, *l′PTi* 7, *l″PTi* 16. Peritreme ending in small expansion (Figure 30D).

Legs (Figure 36). Similar to female except for missing one or two ventral tactile setae and one solenidion, tarsus I with one pair of duplex setae *ω′* 7, and one additional ventral solenidion *vω* 8 long, tibia I with one solenidion, *φ* 9; tarsus II with one solenidion *ω″* 8 long. Segmental length of legs: leg I: trochanter 20, femur 53, genua 30, tibia 28, tarsus 49; leg II: trochanter 16, femur 44, genua 24, tibia 24, tarsus 41; leg III: trochanter 17, femur 33, genua 20, tibia23, tarsus 49; leg IV: trochanter 17, femur 36, genua 20, tibia 22, tarsus 47; legs chaetotaxy I−IV (eupathidia and solenidia in parentheses): trochanters 1-1-1-0; femora 4-3-1-1; genua 2-1-0-0; tibiae 3(0)(1)-2-1-1; tarsi 4(3)(2)-4(3)(1)-5-5.


**Key to species of *Stylophoronychus* (females)**


All dorsal setae club-like, coxal setal count 2-2-1-1...................*S*. *insularis* (Flechtmann)-Most dorsal setae long linear or spatulate, coxal setal count 2-1-1-1.....................2 
Hysterosoma with a central protuberance that arches upward, covered with an irregular pattern of circles or the fusion of many circles...........................*S. vannus* (Rimando)-Hysterosoma arched upward or not and without protuberance..............................3
Length of *c*_1_ and *d*_1_ as long as, or longer than the distances between their respective setal bases and those of the setae in next row.................*S. wangae*
**Pan, Jin & Yi sp. nov.**-Length of *c*_1_ and *d*_1_ less than the distances between their respective setal bases and those of the setae in next row................................................................................4
Setae *c*_1_, *d*_1_ and *e*_1_ decreasing in size successively..................*S. nakaoi* (Ehara & Wongsiri)-Setae *c*_1_, *d*_1_ and *e*_1_ subequal in length...............................................*S. baghensis* (Prasad)


## 4. Discussion

Studying the ontogeny of spider mites will provide a better system for the classification and identification of these species. Several articles have been published to address this issue to reach a better understanding of the ontogenetic development of spider mites [11,12,13,14,15,16,17,18,19,20,21,22]. Previously, all species of *Stylophoronychus* were known only as adults; nothing was known of their ontogenetic development. Here, we discuss the ontogeny of the two species to give a preliminary insight into the evolution of the genus.

Ontogenetic development of *S*. *vannus* is mentioned in the description section above. Compared with the basic pattern of Tetranychinae described by Lindquist [10], the patterns of setation and setal additions on femura to the tarsi of legs I–IV in *Stylophoronychus wangae*
**sp. nov.** have large amounts of setae suppressed.

Trochanters. The pattern and setal additions on the trochanter of legs I–IV in *S. wangae*
**sp. nov.** follow the basic pattern for Tetranychinae described by Lindquist [10]. The seta *v′* is absent on all legs of larval and protonymphal stages and present on deutonymphal trochanters I–III, but not on trochanter IV until reaching the adult stage.

Femora. The formula pattern of deutonymphal legs I–IV in *S. wangae*
**sp. nov.** is 4-3-1-1, bearing three setae *d*, *bv″*, *v′* and *l′* on femur I, three setae *d*, *bv″* and *v′* on femur II, only *d* on femur III and IV. Two setae *l″* and *v″* are suppressed on leg I and *ev′* is suppressed on leg III and leg IV compared with the date (6-3-2-2) of Lindquist [10]. The setal additional patterns only in adult femur I show sexual dimorphism—*l′*_1_ and *l″*_1_ are added to female femur I, while *v″*, *l′*_1_ and *l″*_1_ are added to male femur I; *l′*_1_ and *l″*_1_ are added to adult femur II; *v′* and *l′*_1_ added to adult femur III and none is added on femur IV. In total, six setae and seven setae are added to the femora of the female and male of this species, respectively, during ontogeny. Compared with the normal chaetotaxy of the female described by Lindquist [10], four setae *l″*, *v″*, *v′*_1_ and *v″*_1_ are absent on the female femur I of *S*. *wangae*
**sp. nov.**, *v″*_1_ is suppressed on femur II, and *ev′* is absent on legs III–IV. A total of four setae are suppressed on formula of deutonymphal stage and seven on adult female of this species.

Genua. The genual setation of deutonymphal legs I–IV in *S*. *wangae*
**sp. nov.** is 2-1-0-0, *l′* and *l″* present on genu I, only *l′* on genu II. The basic pattern of deutonymphal genual setation in Tetranychinae as described by Lindquist [10] is 5-5-3-3 in *S*. *wangae*
**sp. nov.**, three setae *d*, *v′* and *v″* are suppressed on genua I, four setae *d*, *l″*, *v′* and *v″* are suppressed on genu II, and genua III and IV each lack three setae (*d*, *l′* and *v′*). Adult female genual setal count is similar to deutonymphal, except seta *l′* is added on adult genua III–IV and the male has one more seta (*v″*) on the genu I and two more setae (*l″* and *v′*) on the genu II than the female. In total, two setae and five setae are added to the genua of the female and male of this species, respectively, during ontogeny. Based on the basic adult genual of 5-5-4-4 in Lindquist [10], a total of 13 setae suppressed on genua of deutonymph and adult female respectively.

Tibiae. The tibial setation of deutonymphal legs I–IV in *S*. *wangae*
**sp. nov.** is 3(1)-2-1-1, with three tactile setae (*db*, *l′* and *l″*) and one solenidion (*φ*) on tibia I, two tactile setae (*d* and *v′*) on tibia II, and seta *d* on tibia III and IV, respectively. No seta is added in adult female; compared to female, two solenidia *φ′* and *φ″* are added on male tibia I and one solenidion (*φ*) is added on tibia II. In total, no setae and three solenidia are added to the tibiae of the female and male of this species, respectively, during ontogeny. Compared with the deutonymphal pattern of Tetranychinae described by Lindquist [10], four tactile setae (*l′*_1_, *l″*_1_, *l′* and *l″*) suppressed on tibia I of *S*. *wangae*
**sp. nov.**, three tactile setae (*l″*, *l′* and *v″*) suppressed on tibia II, and four tactile setae (*d*, *l″*, *l′* and *v″*) suppressed on tibia III and IV, respectively. In the female adult, six tactile setae (*l′*, *l″*, *l′*_1_, *l″*_1_, *v′*_1_ and *v″*_1_) suppressed on tibia I, five tactile setae (*l″*, *v′*, *v″*, *l′*_1_ and *v′*_1_) suppressed on tibia II, five tactile setae (*d*, *l″*, *l′*, *v″* and *v′*_1_) suppressed on tibia III, six tactile setae (*d*, *l″*, *v′*, *v″*, *l′*_1_ and *v′*_1_) suppressed on tibia IV. A total of 15 setae are suppressed on tibiae of deutonymphal stage and 22 on adult female of this species.

Tarsi. All stages lack one ventral seta *pv″,* setae *tc* and *ft* unpaired on tarsi I–II, only present one seta *tc′* but no *tc″* and one fastigial seta *ft′* but not *ft″*. In the deutonymph, four tactile setae (*u′*, *u″*, *ft′* and *tc′*), three eupathidia (*p′ζ*, *p″ζ* and *pv′ζ*) and two solenidia(*ω′* and *vω*) present on leg I; four tactile setae (*u′*, *u″*, *ft′* and *tc′*), three eupathidia (*p′ζ*, *p″ζ* and *pv′ζ*) and one solenidion (*ω″*) on leg II, five tactile setae (*u′*, *u″*, *ft′*, *ft″* and *pv′*) present on tarsus III–IV respectively. Adult female tarsal setae are similar to deutonymphal, except *v′*_1_ and *v″*_1_ are added on tarsi I–II, *v″*_1_ and *ω′* are added on tarsi III–IV and the male added one more seta (*ω″*_1_) than the female on tarsi I–II. In total, no setae and three solenidia are added to the tarsi of the female and male of this species, respectively, during ontogeny.

Compared with the basic pattern of Tetranychinae described by Lindquist [10], it appears that six tactile setae (*ft″*, *tc″*, *l′*, *l″*, *v′*_2_ and *pv″*) and one solenidion (*ω″*_1_) are suppressed on deutonymphal leg I of *S*. *wangae*
**sp. nov.**, four tactile setae (*ft″*, *tc″*, *v″*_1_ and *pv″*) suppressed on leg II, three tactile setae (*tc′*, *tc″* and *pv″*) and one solenidion (*ω′*) suppressed on leg III, three tactile setae (*tc′*, *tc″* and *pv″*) are suppressed on leg IV and, in total, 18 setae are suppressed on deutonymphal legs I–IV in *S*. *wangae*
**sp. nov.** In the adult female, seven setae (*ft″*, *tc″*, *l′*, *l″*, *v′*_2_, *pv″* and *ω′*) are suppressed on tarsus I; six setae (*ft″*, *tc″*, *l′*, *v′*_2_, *pv″* and *ω″*_1_) are suppressed on tarsus II; three setae (*tc′*, *tc″* and *pv″*) are suppressed on tarsus III; four setae (*tc′*, *tc″*, *v′*_1_ and *pv″*) are suppressed on tarsus IV. In total, 20 setae are suppressed on adult female legs I–IV in *S*. *wangae*
**sp. nov**.

Compared with the data presented by Lindquist [10] for the Tetranychidae, there are several differences during the ontogeny of two species of *Stylophoronychus*, as follows: 

(1)Two additional setae, ventral solenidion (*vω*) on tarsus I and ventral seta (*v″*_1_) on tarsus III in male of *S. vannus* and *S. wangae*
**sp. nov.**, do not express setal standard for the Tetranychidae;(2)Unpaired tectal seta (*tc′*) and fastigial seta (*ft′*) present on tarsi I and II, and paired are *tc* suppressed on tarsus III and IV in two species and unpaired ventral seta *pv″* presents in *S*. *wangae*
**sp. nov.**;(3)Seta *ev′* is suppressed on femur III and IV;(4)Seta *v′* on trochanter IV is suppressed in deutonymph and delayed additions in adult.

## Figures and Tables

**Figure 1 insects-13-01176-f001:**
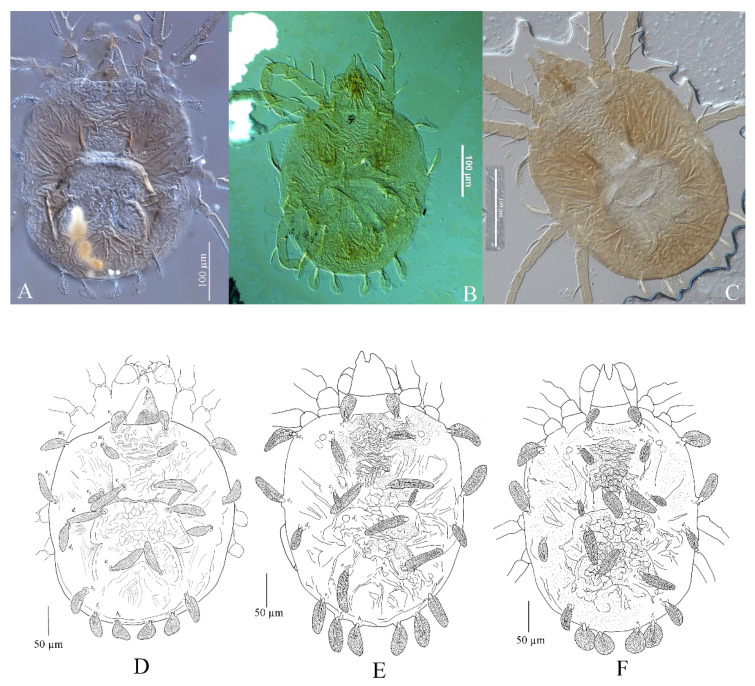
Idiosoma of adult females. *S*. *guangzhouensis* (**A**,**D**); *S*. *lalli* (**B**,**E**); and *S. vannus* (**C**,**F**).

**Figure 2 insects-13-01176-f002:**
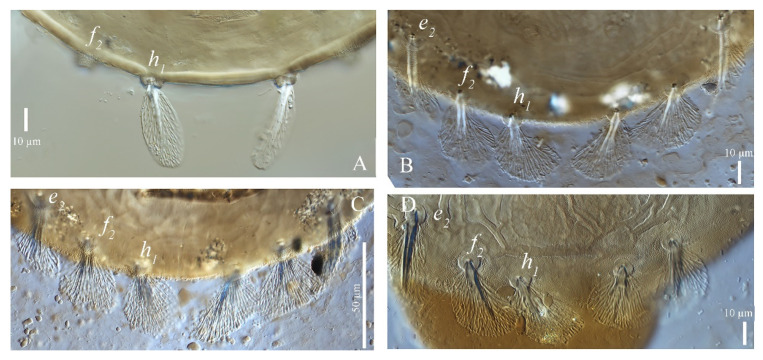
Female, variations of caudal and sacral setae (*e*_2_, *f*_2_, *h*_1_) of *S*. *guangzhouensis* (**A**–**D**).

**Figure 3 insects-13-01176-f003:**
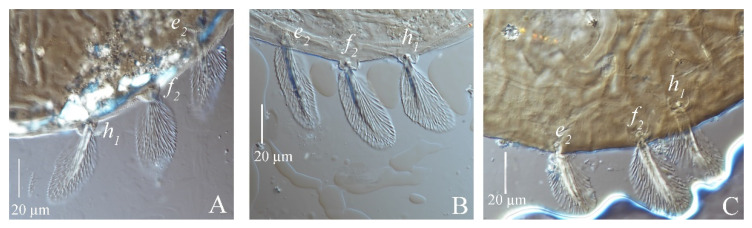
Female, caudal and sacral setae (*e*_2_, *f*_2_, *h*_1_). *S. guangzhouensis* (**A**); *S. lalli* (**B**); and *S. vannus* (**C**).

**Figure 4 insects-13-01176-f004:**
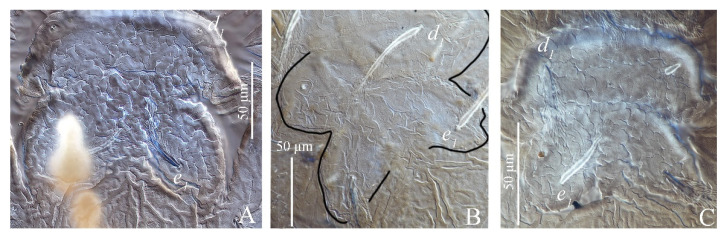
Patterns of hysterosomal median protuberance of adult female of three species of *Stylophoronychus*. *S*. *guangzhouensis* (**A**); *S*. *lalli* (**B**); and *S. vannus* (**C**). “Black line” in Figure 4B refer to the outline shape of protuberance. “Black line” in Figure 4B refer to the shape of the hysterosomal median protuberance of *S*. *lalli* female, because the outline of the photo is not clear.

**Figure 5 insects-13-01176-f005:**
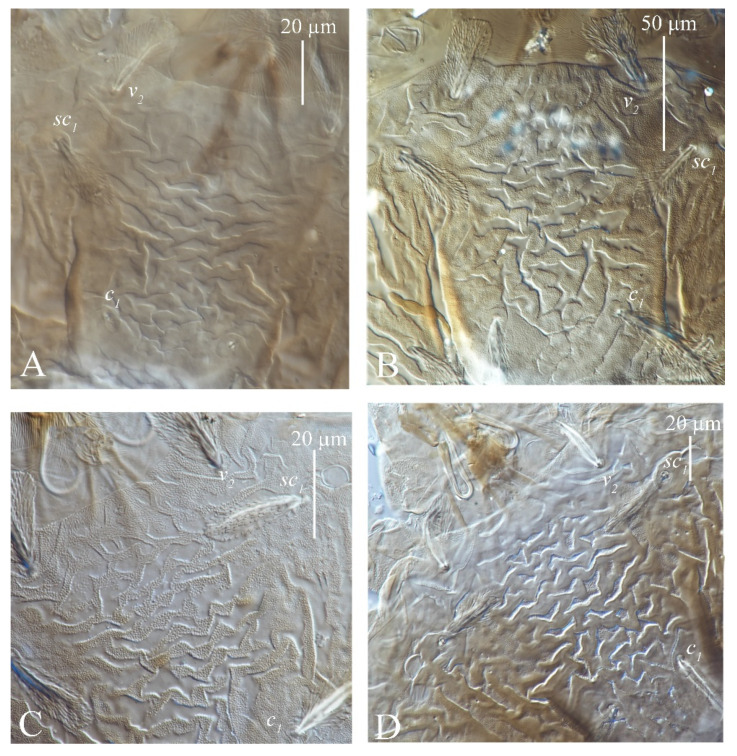
Patterns of integument on medial prodorsum of adult female of three species of *Stylophoronychus*. *S. guangzhouensis* (**A**,**B**); *S. lalli* (**C**); and *S. vannus* (**D**).

**Figure 6 insects-13-01176-f006:**
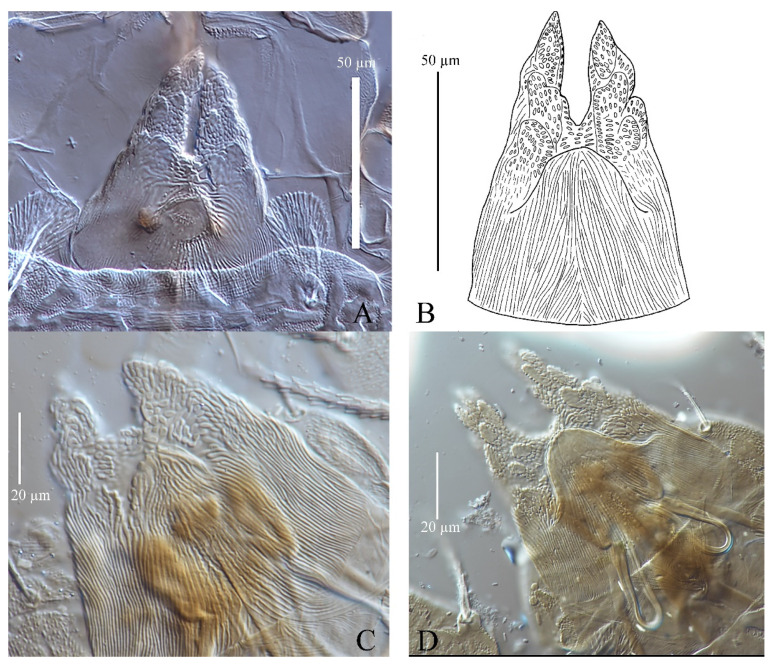
Adult female stylophore of three species of *Stylophoronychus*. *S*. *guangzhouensis* (**A**,**B**); *S*. *lalli* (**C**); and *S. vannus* (**D**).

**Figure 7 insects-13-01176-f007:**
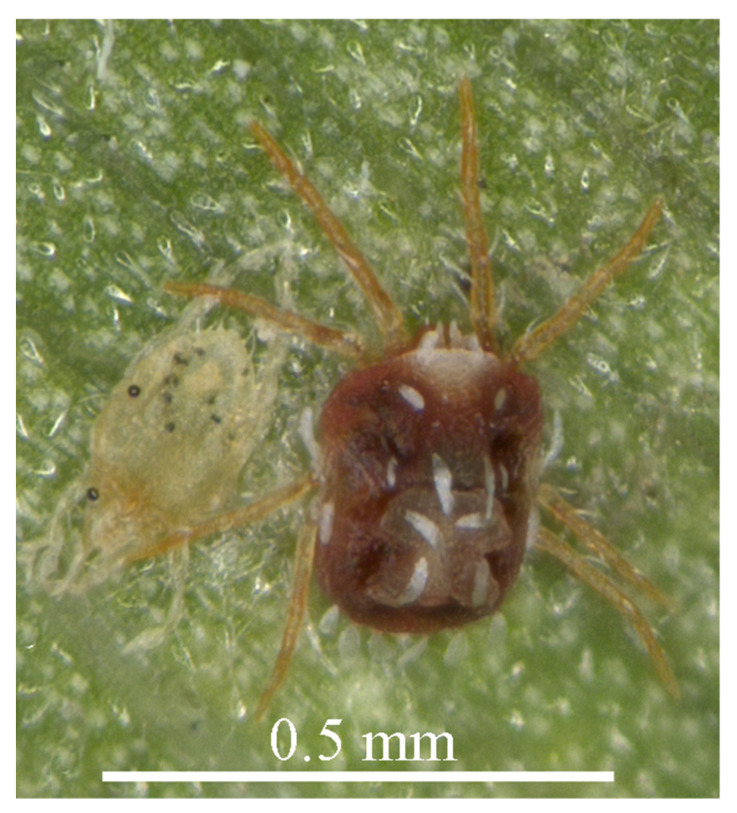
*Stylophoronychus vannus* (Rimando, 1968). Photograph. Female on a leaf of bamboo.

**Figure 8 insects-13-01176-f008:**
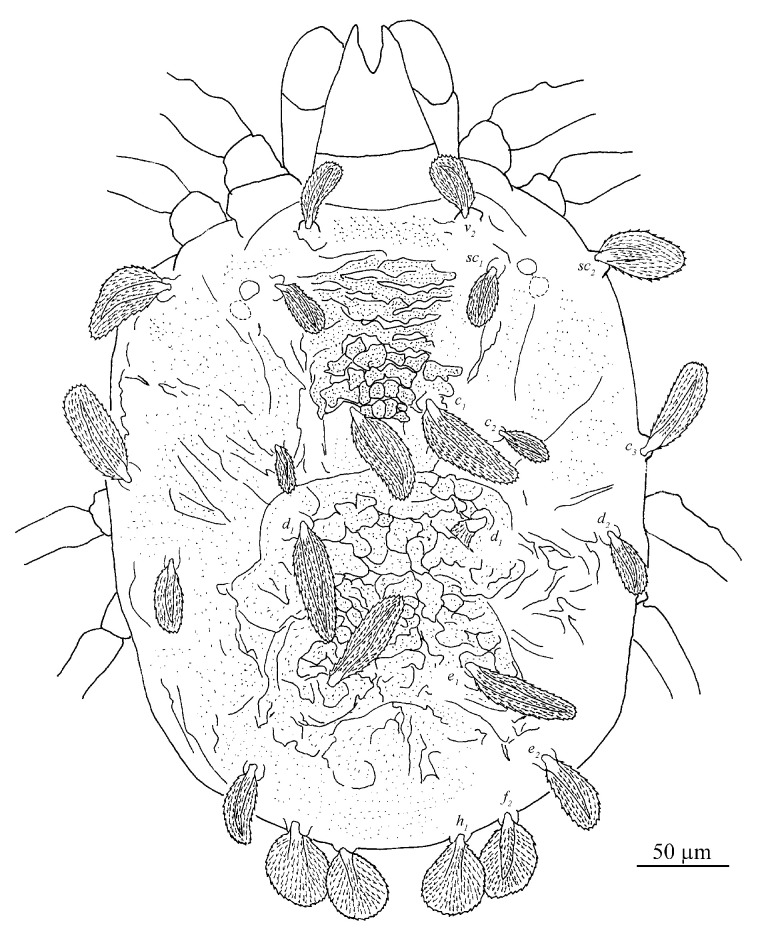
*Stylophoronychus vannus* (Rimando, 1968). Female: dorsal view of idiosoma.

**Figure 9 insects-13-01176-f009:**
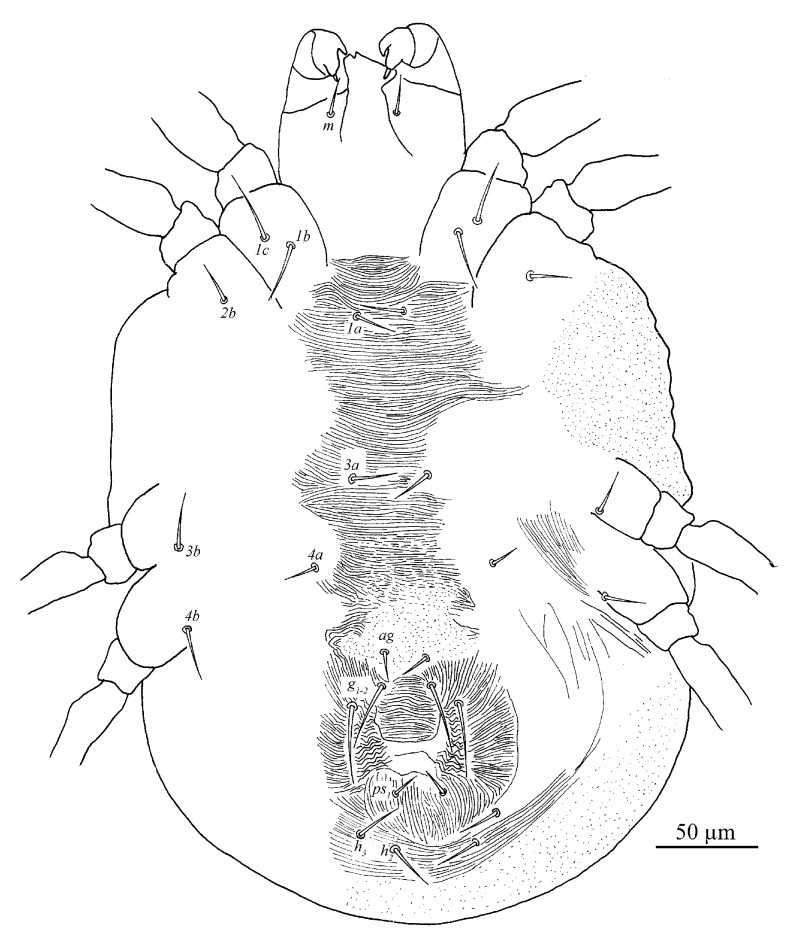
*Stylophoronychus vannus* (Rimando, 1968). Female: ventral view of idiosoma.

**Figure 10 insects-13-01176-f010:**
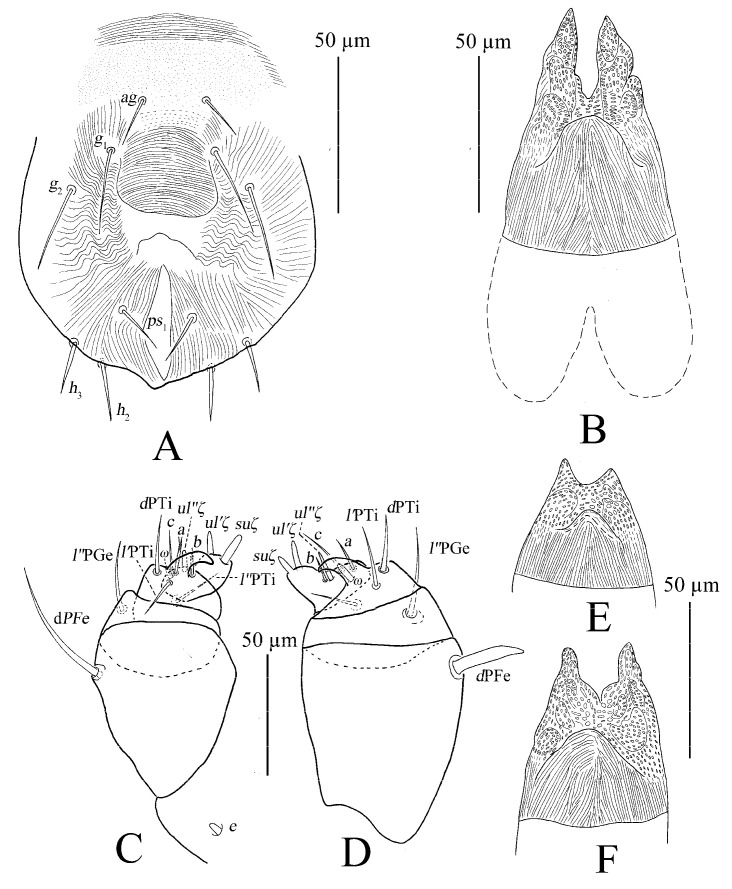
*Stylophoronychus vannus* (Rimando, 1968). (**A**) Female: genital and anal region; (**B**) female: stylophore; (**C**) female: femur, genu, tibia and tarsus of palp; (**D**) male: femur, genu, tibia and tarsus of palp; (**E**) male: stylophore; and (**F**) deutonymph: stylophore.

**Figure 11 insects-13-01176-f011:**
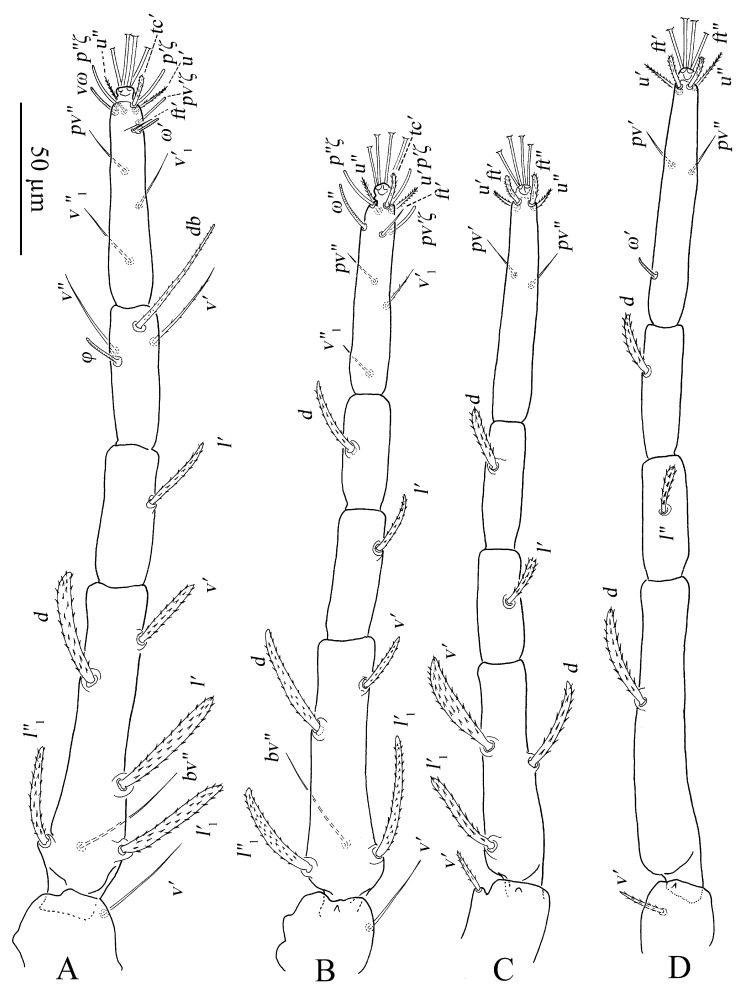
*Stylophoronychus vannus* (Rimando, 1968). Female: (**A**–**D**) trochanter–tarsus of legs I–IV, respectively.

**Figure 12 insects-13-01176-f012:**
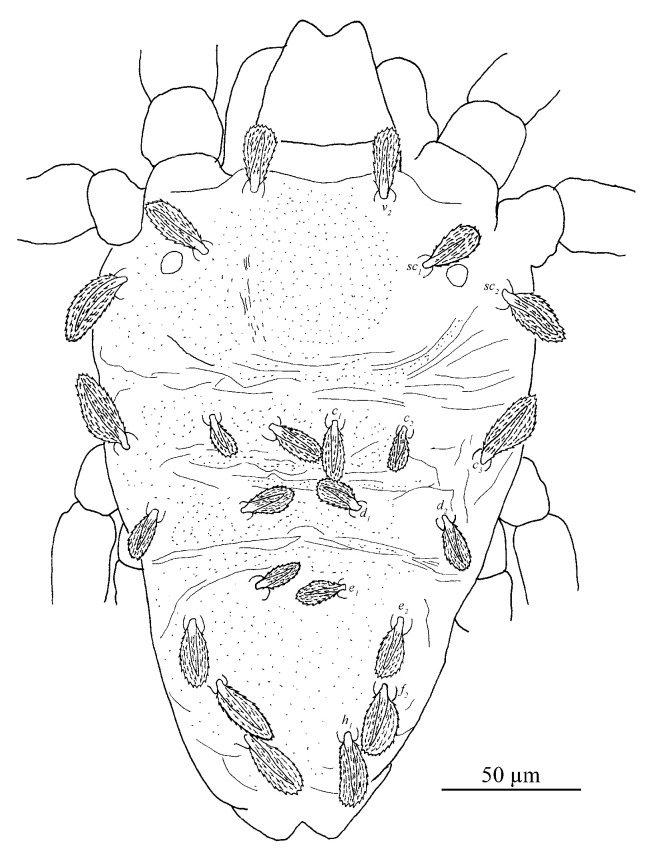
*Stylophoronychus vannus* (Rimando, 1968). Male: dorsal view of idiosoma.

**Figure 13 insects-13-01176-f013:**
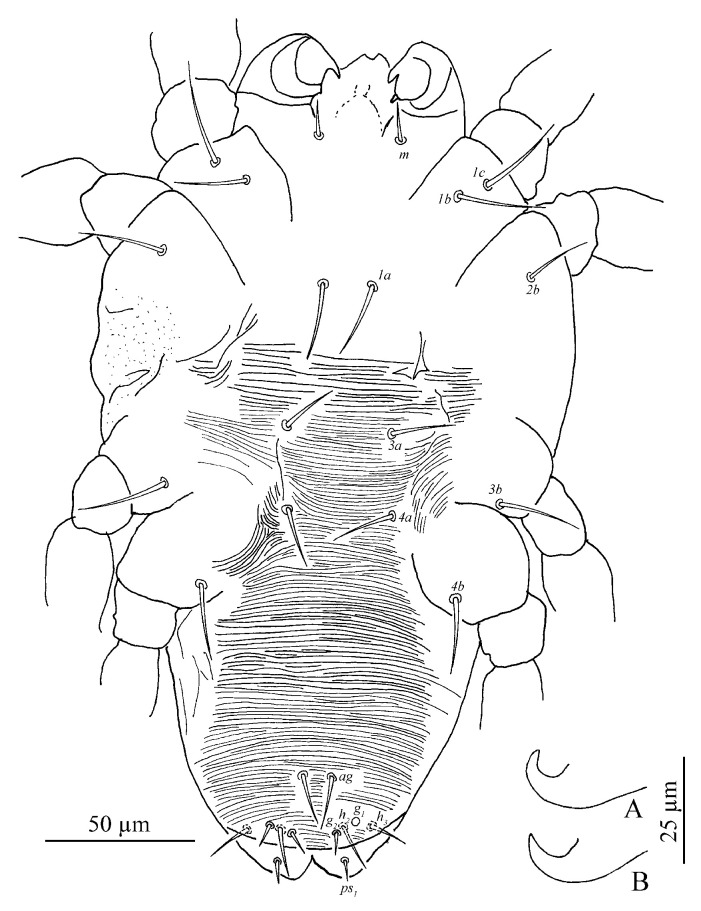
*Stylophoronychus vannus* (Rimando, 1968). Male: ventral view of idiosoma. (**A**,**B**) aedeagus.

**Figure 14 insects-13-01176-f014:**
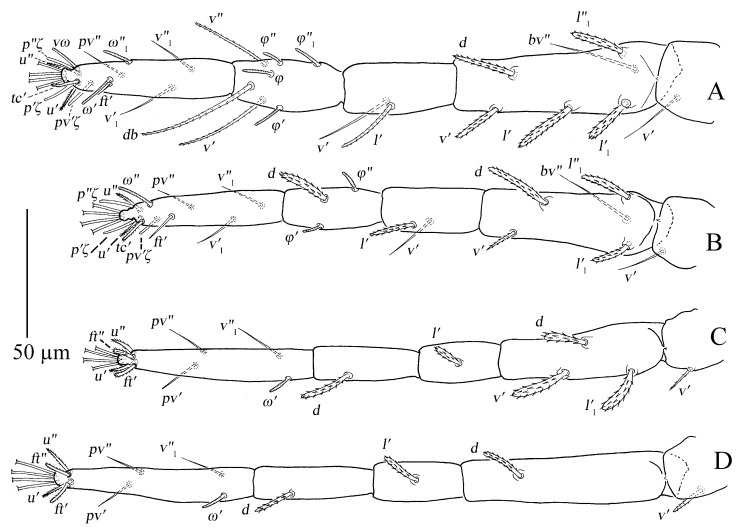
*Stylophoronychus vannus* (Rimando, 1968). Male: (**A**–**D**) trochanter–tarsus of legs I–IV, respectively.

**Figure 15 insects-13-01176-f015:**
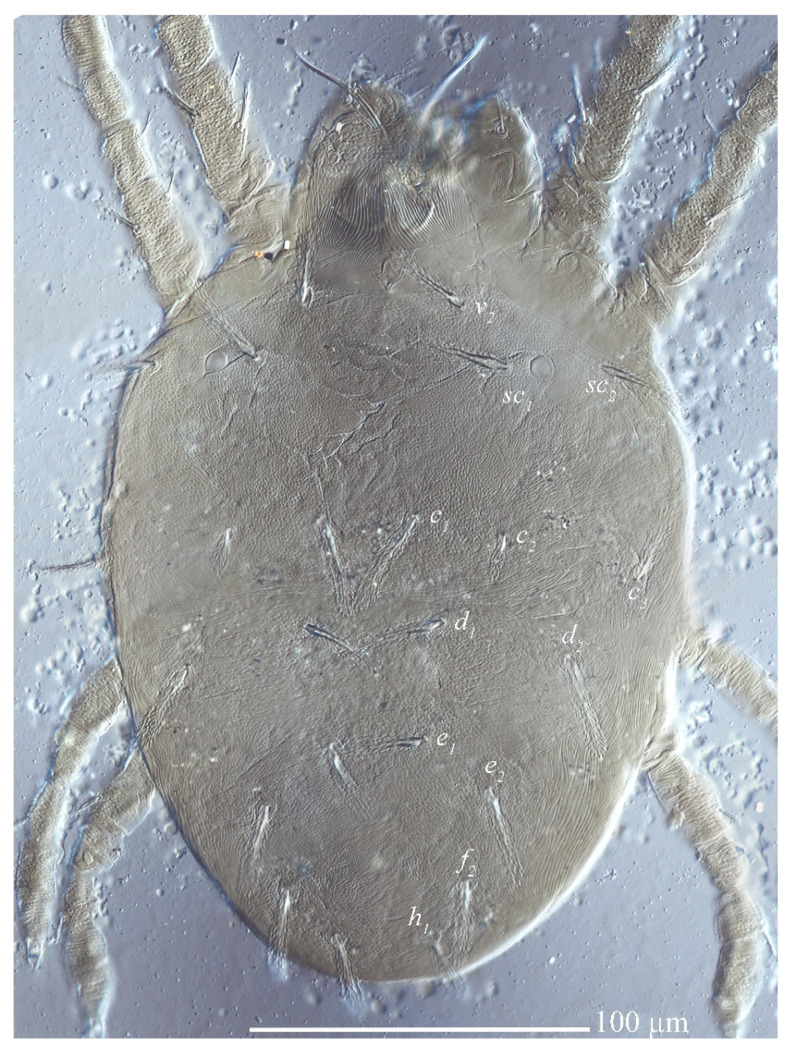
*Stylophoronychus vannus* (Rimando, 1968). Photograph. Deutonymph: dorsal view of idiosoma.

**Figure 16 insects-13-01176-f016:**
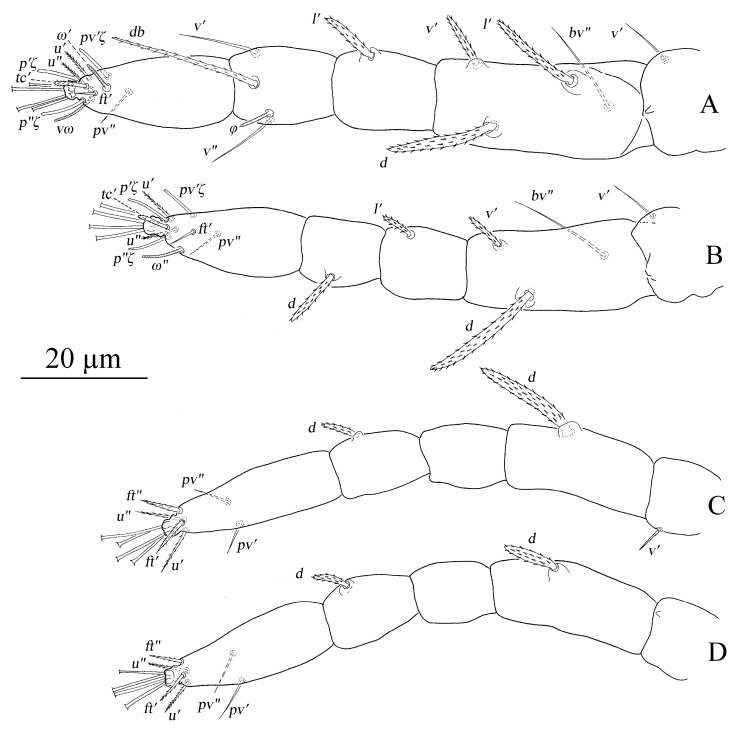
*Stylophoronychus vannus* (Rimando, 1968). Deutonymph: (**A**–**D**) trochanter–tarsus of legs I–IV, respectively.

**Figure 17 insects-13-01176-f017:**
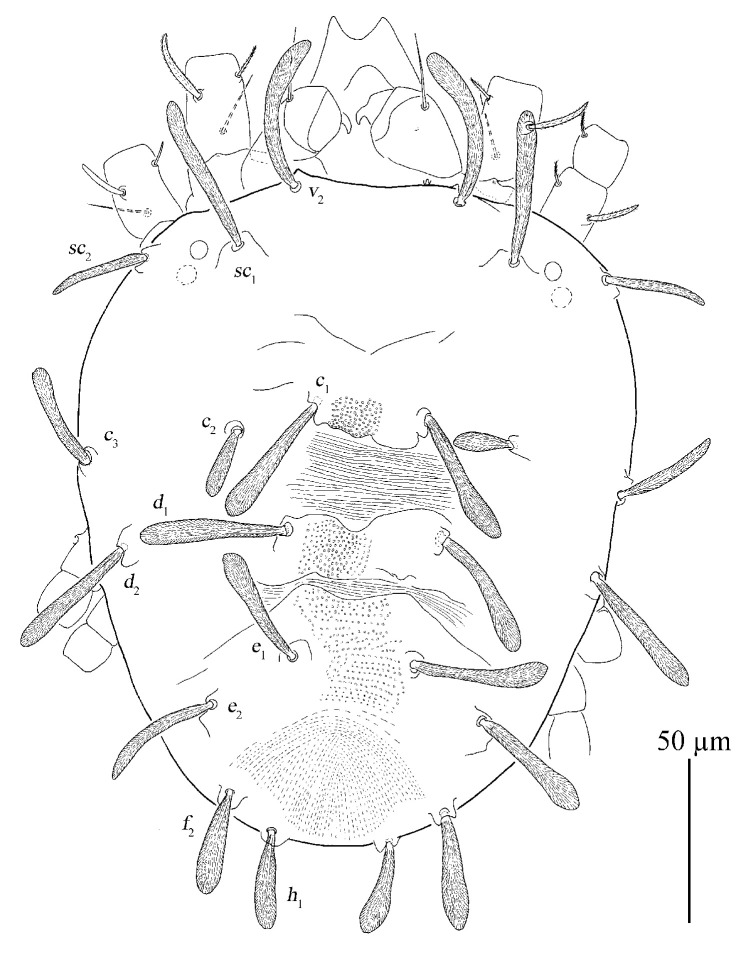
*Stylophoronychus vannus* (Rimando, 1968). Protonymph: dorsal view of idiosoma.

**Figure 18 insects-13-01176-f018:**
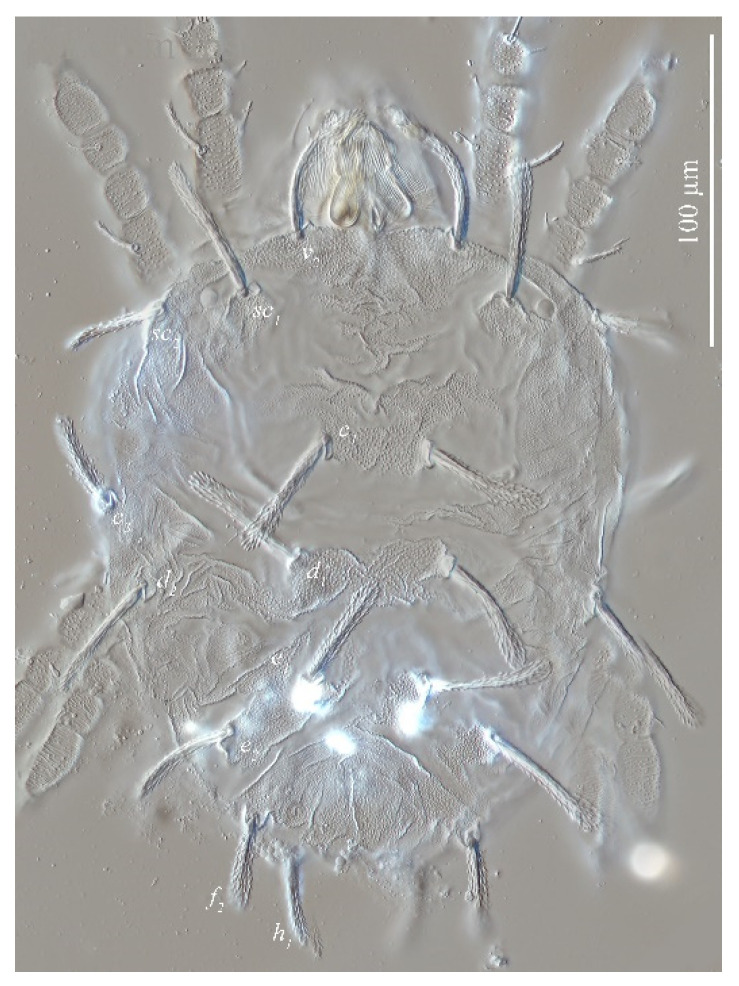
*Stylophoronychus vannus* (Rimando, 1968). Photograph. Protonymph: dorsal view of idiosoma.

**Figure 19 insects-13-01176-f019:**
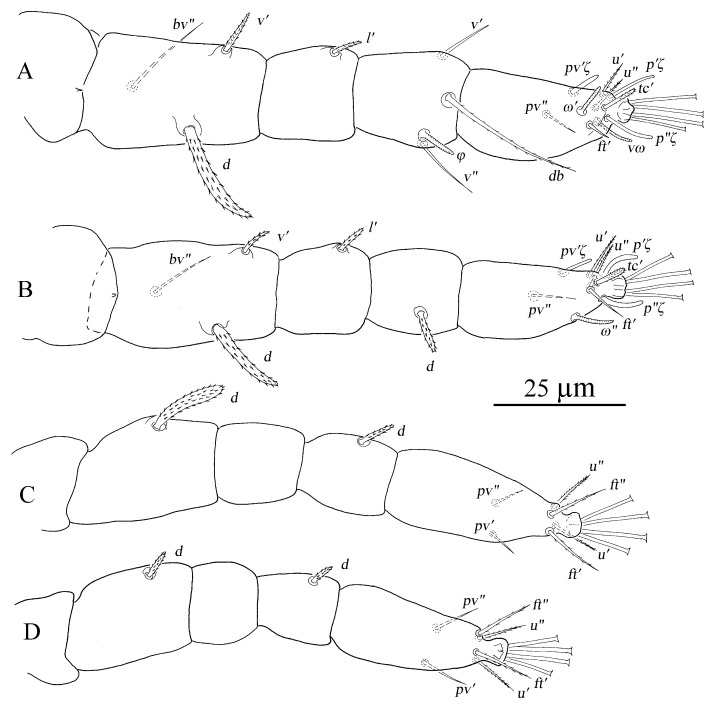
*Stylophoronychus vannus* (Rimando, 1968). Protonymph: (**A**–**D**) trochanter–tarsus of legs I–IV, respectively.

**Figure 20 insects-13-01176-f020:**
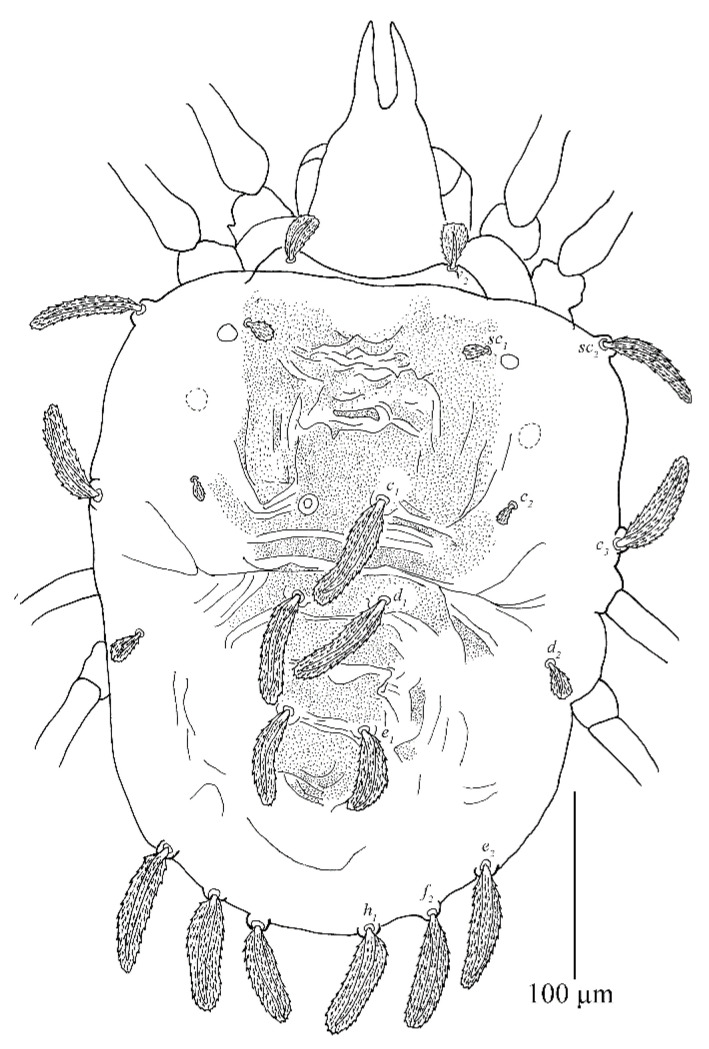
*Stylophoronychus wangae***Pan, Jin & Yi sp. nov.** Adult female: dorsal view of idiosoma.

**Figure 21 insects-13-01176-f021:**
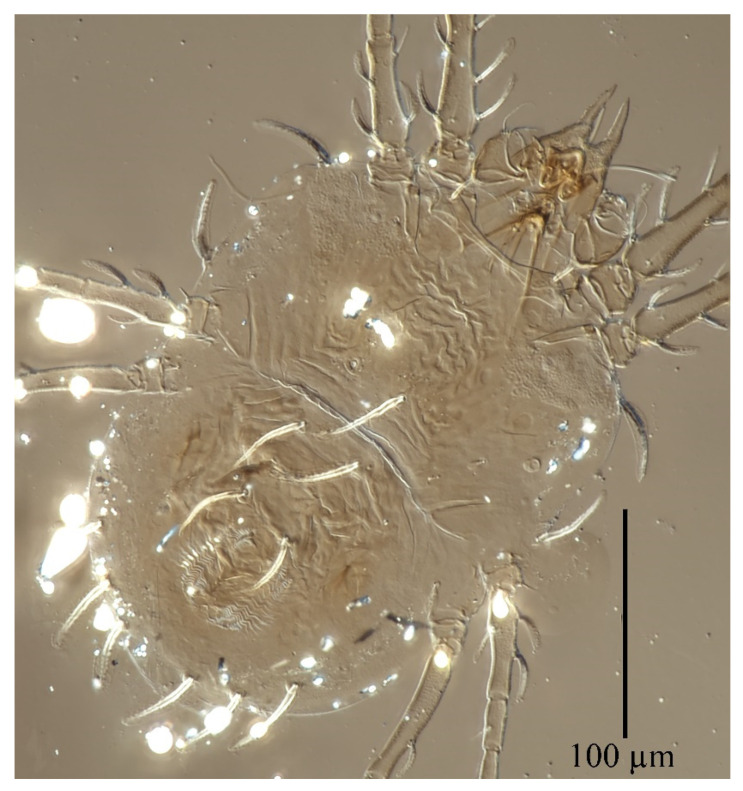
*Stylophoronychus wangae***Pan, Jin & Yi sp. nov.** Photograph. Adult female: dorsal view of idiosoma.

**Figure 22 insects-13-01176-f022:**
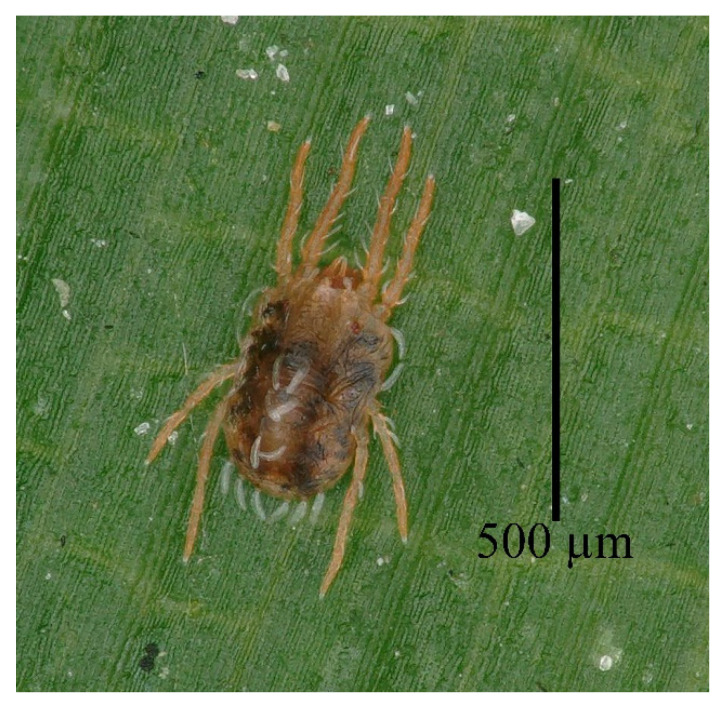
*Stylophoronychus wangae***Pan, Jin & Yi sp. nov.** Photograph. Female on leaf of bamboo.

**Figure 23 insects-13-01176-f023:**
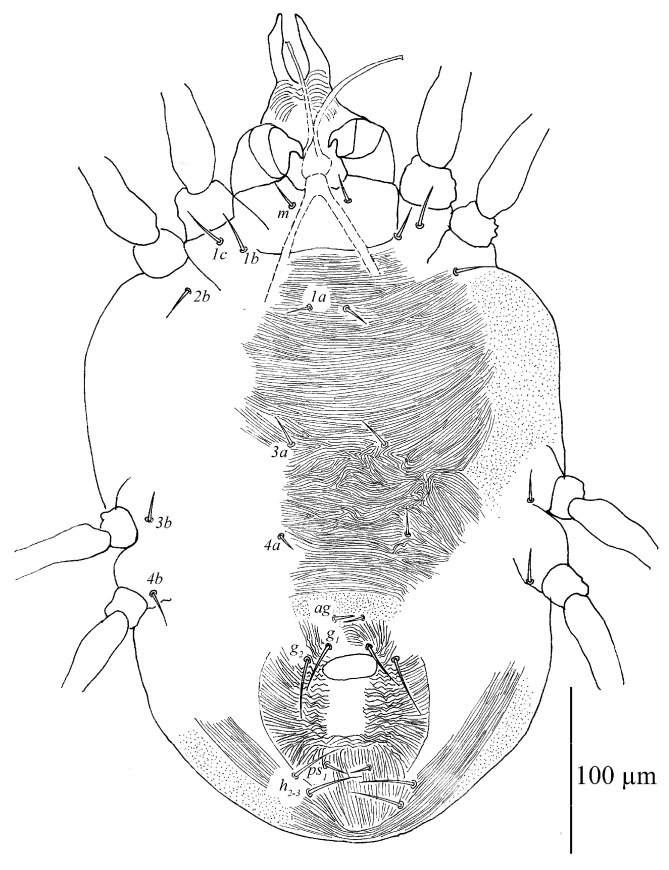
*Stylophoronychus wangae***Pan, Jin & Yi sp. nov.** Female: ventral view of idiosoma.

**Figure 24 insects-13-01176-f024:**
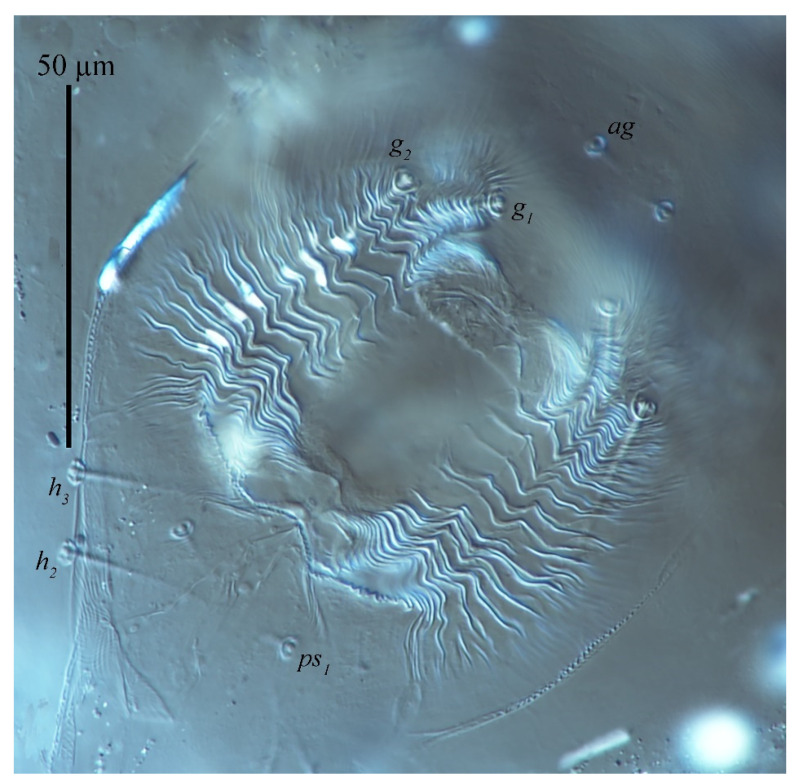
*Stylophoronychus wangae***Pan, Jin & Yi sp. nov.** Photograph. Female: genital and anal region.

**Figure 25 insects-13-01176-f025:**
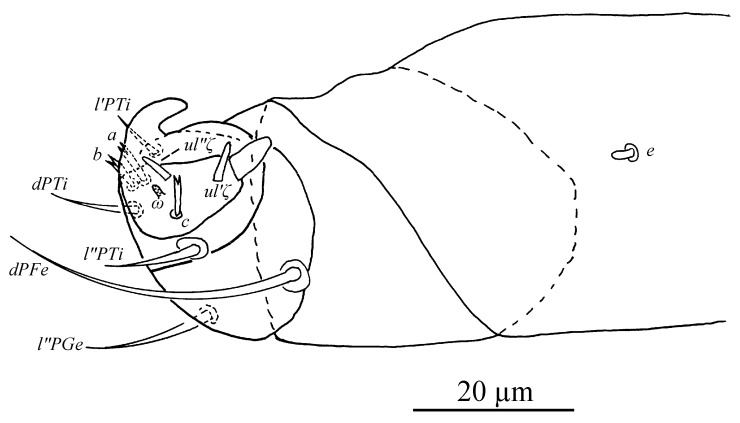
*Stylophoronychus wangae***Pan, Jin & Yi sp. nov.** Female: femur, genu, tibia and tarsus of palp.

**Figure 26 insects-13-01176-f026:**
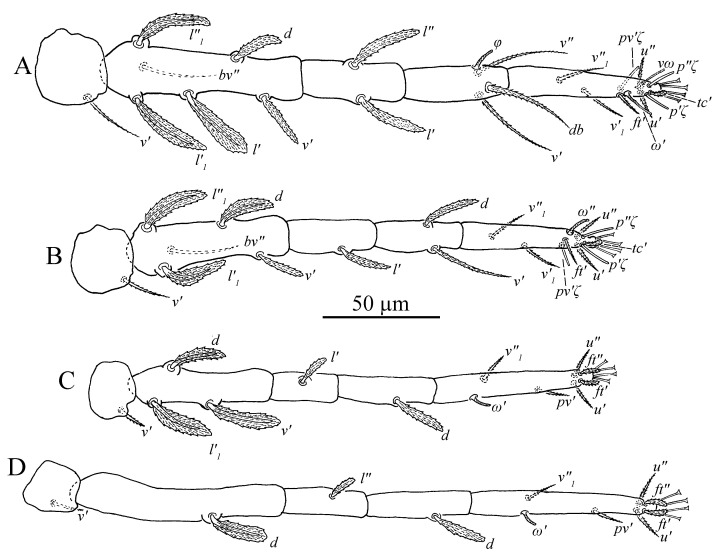
*Stylophoronychus wangae***Pan, Jin & Yi sp. nov.** Female: (**A**–**D**) trochanter–tarsus of legs I–IV, respectively.

**Figure 27 insects-13-01176-f027:**
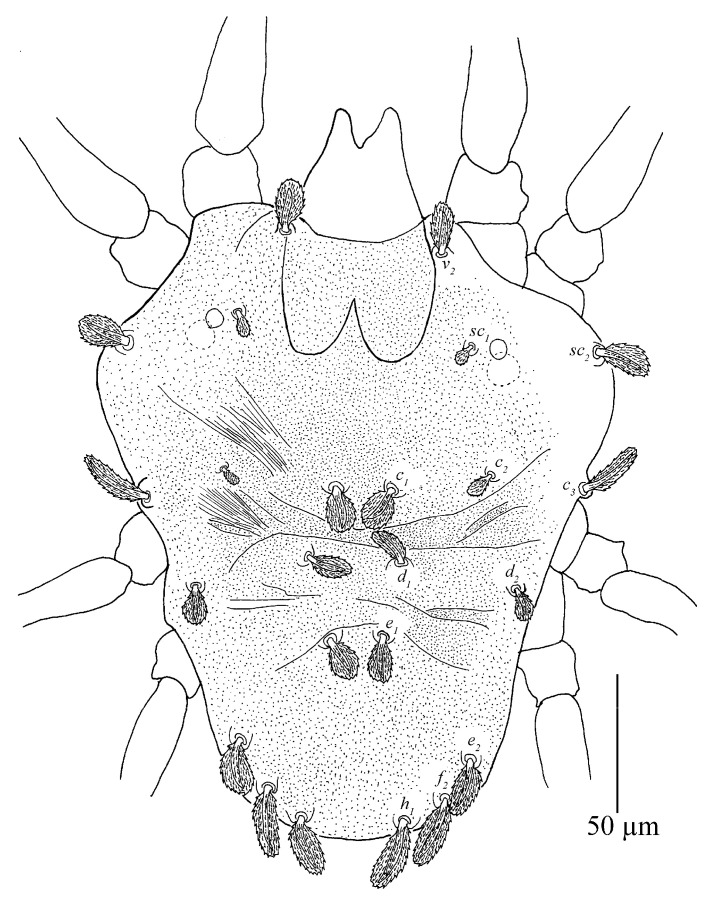
*Stylophoronychus wangae***Pan, Jin & Yi sp. nov.** Male: dorsal view of idiosoma.

**Figure 28 insects-13-01176-f028:**
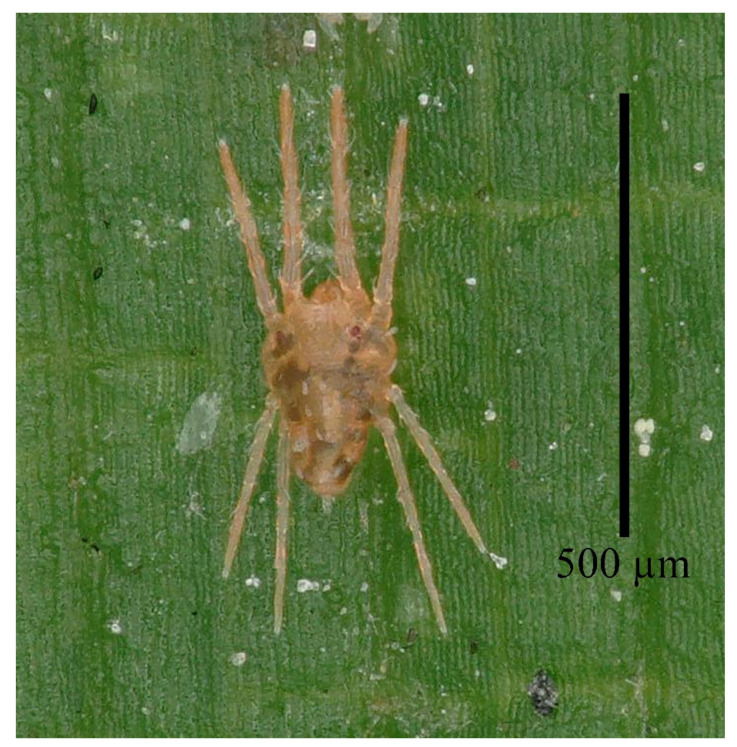
*Stylophoronychus wangae***Pan, Jin & Yi sp. nov.** Photograph. Male on leaf of bamboo.

**Figure 29 insects-13-01176-f029:**
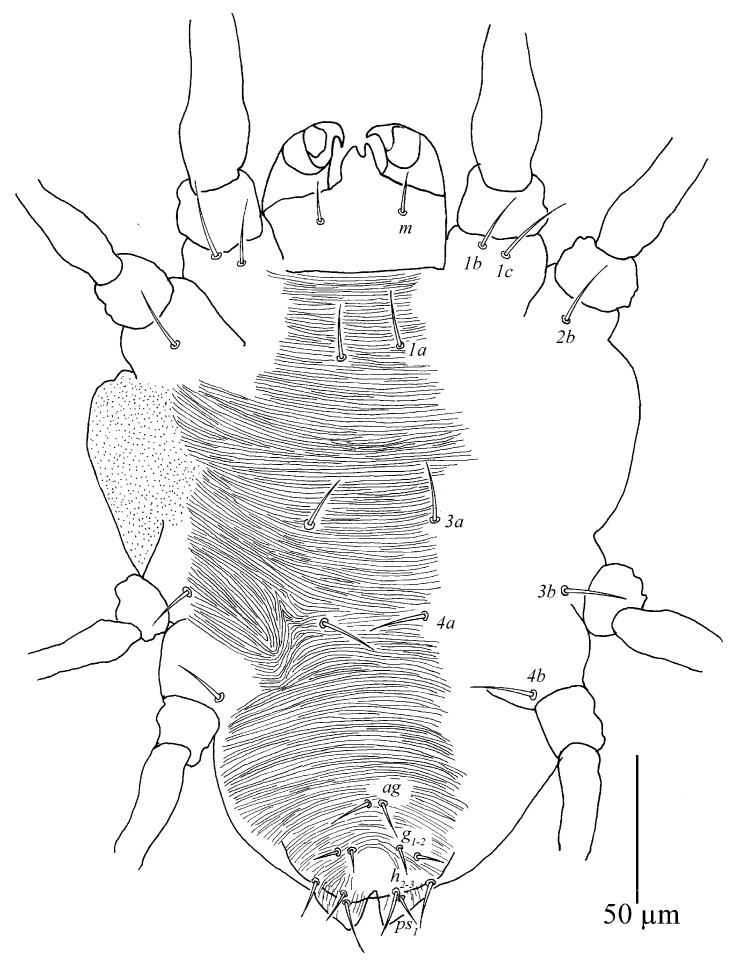
*Stylophoronychus wangae***Pan, Jin & Yi sp. nov.** Male: ventral view of idiosoma.

**Figure 30 insects-13-01176-f030:**
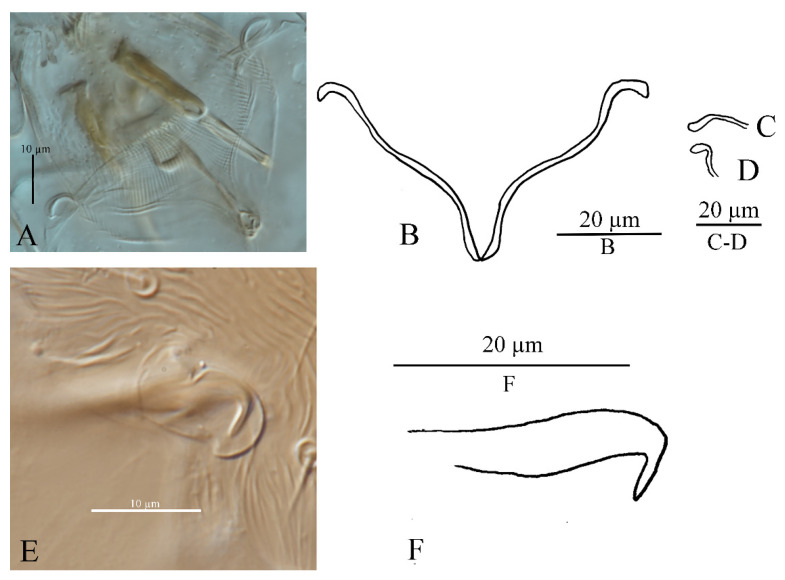
*Stylophoronychus wangae***Pan, Jin & Yi sp. nov.** (**A**,**B**) Male peritreme; (**C**,**D**) apical part of peritreme of female and deutonymph, respectively; and (**E**,**F**) aedeagus.

**Figure 31 insects-13-01176-f031:**
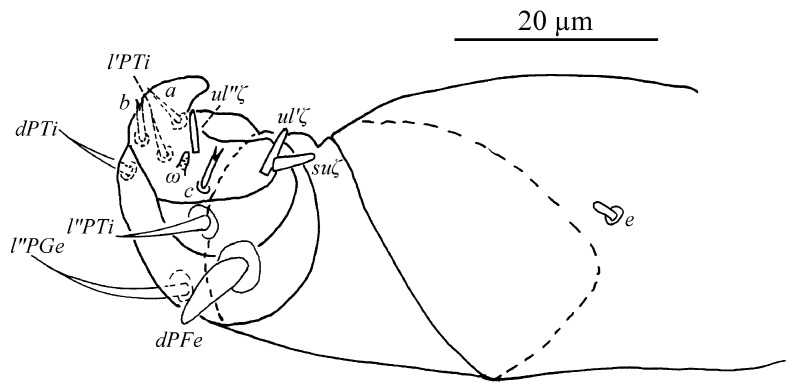
*Stylophoronychus wangae***Pan, Jin & Yi sp. nov.** Male: femur, genu, tibia and tarsus of palp.

**Figure 32 insects-13-01176-f032:**
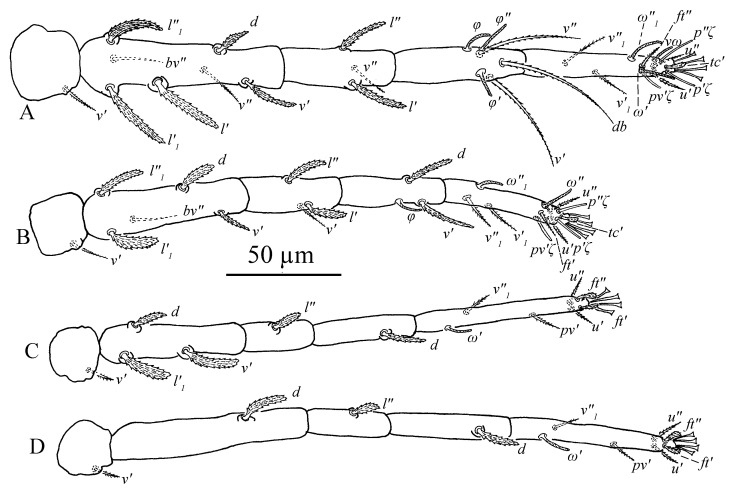
*Stylophoronychus wangae***Pan, Jin & Yi sp. nov.** Male: (**A**–**D**) trochanter–tarsus of legs I–IV, respectively.

**Figure 33 insects-13-01176-f033:**
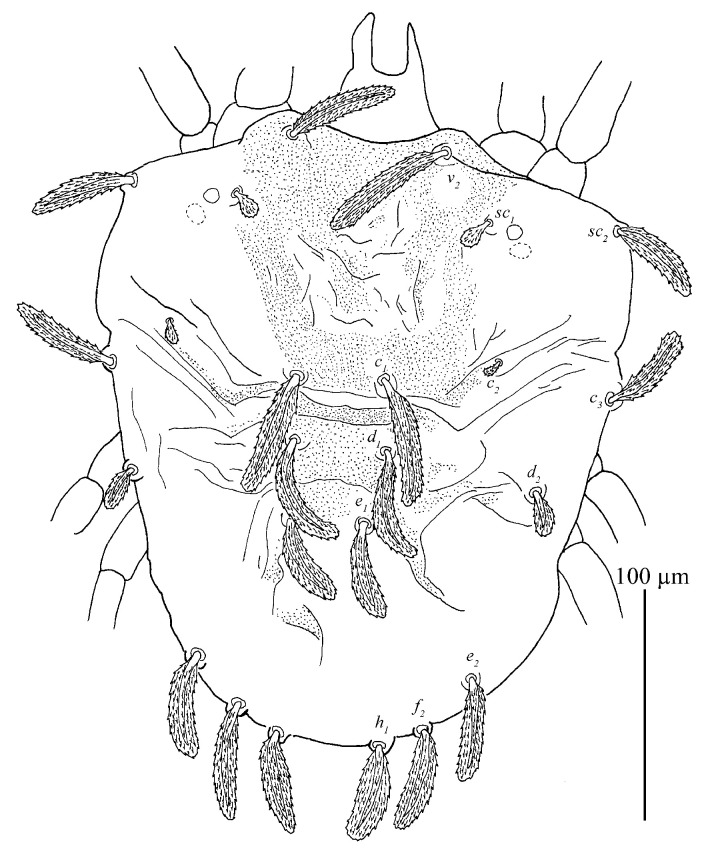
*Stylophoronychus wangae***Pan, Jin & Yi sp. nov.** Deutonymph: dorsal view of idiosoma.

**Figure 34 insects-13-01176-f034:**
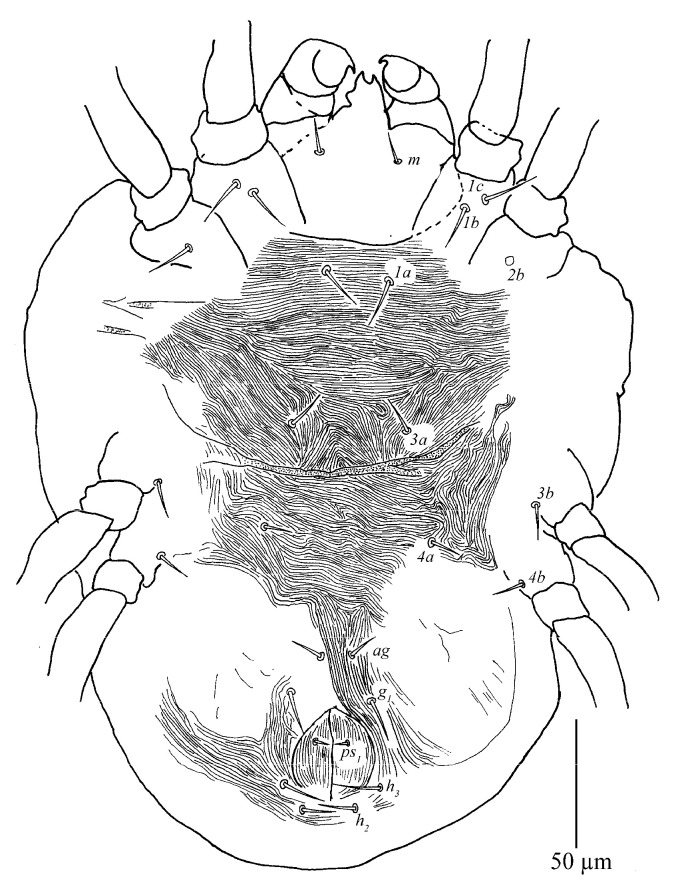
*Stylophoronychus wangae***Pan, Jin & Yi sp. nov.** Deutonymph: ventral view of idiosoma.

**Figure 35 insects-13-01176-f035:**
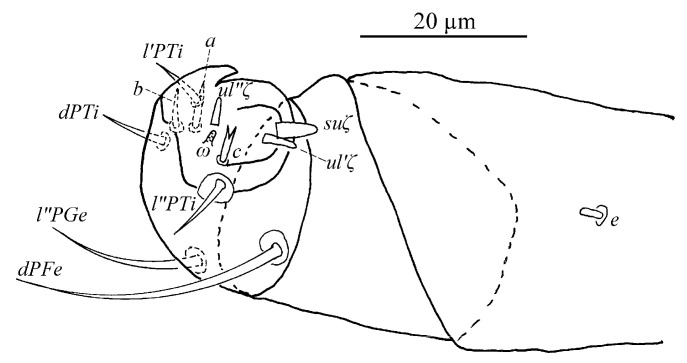
*Stylophoronychus wangae***Pan, Jin & Yi sp. nov.** Deutonymph: femur, genu, tibia and tarsus of palp.

**Figure 36 insects-13-01176-f036:**
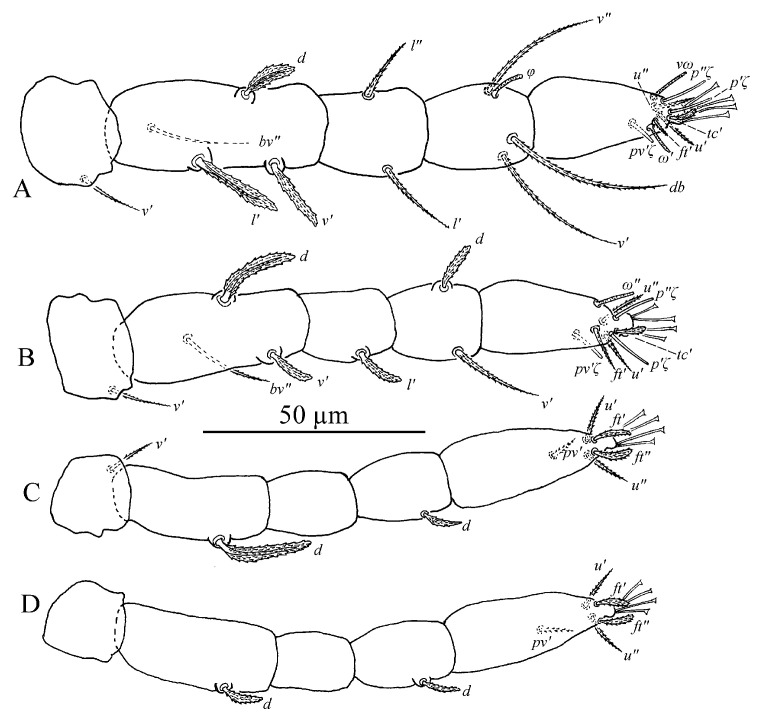
*Stylophoronychus wangae***Pan, Jin & Yi sp. nov.** Deutonymph: (**A**–**D**) trochanter–tarsus of legs I–IV, respectively.

**Table 1 insects-13-01176-t001:** Measurements of idiosoma and dorsal setae of three species of *Stylophoronychus*.

Species (Female)	*S. guangzhouensis* *n* = 19	*S. lalii* *n* = 1	*S. vannus* *n = 3*
Idiosoma			
Length (*v*_2_–*h*_1_)	299–338	296	311–324
Width (*c*_3_–*c*_3_)	269–297	264	272–281
Dorsal setae			
*v* _2_	31–38	35	34–39
*sc* _1_	32–37	40	30–38
*sc* _2_	50–56	55	44–55
*c* _1_	62–72	62	59–78
*c*_1_–*c*_1_	48–53	39	38–45
*c*_1_–*d*_1_	58–62	58	59–76
*c* _2_	22–29	26	24–29
*c* _3_	50–65	55	56–61
*d* _1_	61–68	74	60–76
*d*_1_–*d*_1_	85–100	96	95–100
*d*_1_–*e*_1_	76–85	78	79–91
*d* _2_	41–51	38	34–44
*e* _1_	54–72	68	60–66
*e*_1_–*e*_1_	59–90	83	74–80
*e*_1_–*f*_2_	63–82	68	59–86
*e* _2_	43–52	53	43–48
*f* _2_	41–46	50	39–47
*h* _1_	37–46	48	39–42

**Table 2 insects-13-01176-t002:** The number of mutations on tarsus III–IV of *Stylophoronychus vannus* (solenidia in parentheses).

Type	Left Legs	Right Legs	Proportion
1	6-6(1)	6-6(1)	10/23
2	6-6(1)	7-6(1)	3/23
3	6-6(1)	6(1)-6(1)	1/23
4	6-7	6(1)-6(1)	1/23
5	6-7(1)	6-6(1)	1/23
6	7-6(1)	7-6(1)	1/23
7	7-7(1)	7-6(1)	1/23
8	7-7(1)	7-7(1)	4/23
9	7-6(1)	Leg III broken	1/23

## Data Availability

All data are available in this paper.

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
