# Peer review of "Review on the Genus Stylophoronychus (Acari: Tetranychidae), with Description of a New Speciesâ€"

_insects, 2022, doi:10.3390/insects13121176_

Round 1
Reviewer 1 Report
Line 59 ., delete ,
Line 90 insert is is approximately equal . . .
Line 132 insert space besynonymous
Line 156 insert on on those .....
Line 193 duplex setae or associate setae ? compare with line 485
Line 306 delete more
Line 310 ..additions - v' on Genua ??
Line 337 one pair not one pairs 2 times
Line 344 one pair not pairs
Line 360 ...respectiverly. = ?? delete =
Line 393 one pair
Line 440 with convex that bears or with convex bulge that bears ?
Line 477 elongate, blunt tipped delete with
Line 485 Two solenidia compare with line 591
Line 533 + 539 Figure F the drawing shows clearly that the aedeagus is upturned and not downturned ! the shorter line of the shaft is dorsal !
Line 623 insert space adultstage
Lines 724-725 South Africa Department of Agriculture and Water Supply, Entomology Memoir 69
is the correct reference.
Reviewer 2 Report
My comments concern paper „Review on the genus Stylophoronychus (Acari: Tetranychidae), with description of a new species of Pan et al.”.
It is a very careful, laborious and important paper and in my opinion should be published in Insects.
We get a lot of interesting information in taxonomic discussion about the species of the genus Stylophoronychus and redescription of one species and description of one species.
My only comments: My comments concern paper „Review on the genus Stylophoronychus (Acari: Tetranychidae),It is worth specifying in the paper from which geographical region the palaearctic or oriental come the discussed species, including new sp. In the “Introduction”, it would be better to use the zoogeographical term instead of Asia. What does "small genus" mean in “Summary” ? Wouldn't body height be important in redescription and description besides length and width ?
